# Red List assessment of amphibian species of Ecuador: A multidimensional approach for their conservation

H. Mauricio Ortega-Andrade[1,2]*, Marina Rodes Blanco[1], Diego F. Cisneros-Heredia[2,3], Nereida Guerra Arévalo[1], Karima Gabriela López de Vargas-Machuca[4,5], Juan C. Sánchez-Nivicela[2,3,6], Diego Armijos-Ojeda[7], José Francisco Cáceres Andrade[8], Carolina Reyes-Puig[2,3], Amanda Belén Quezada Riera[9], Paul Székely[7], Octavio R. Rojas Soto[10], Diana Székely[7], Juan M. Guayasamin[11], Fausto Rodrigo Siavichay Pesántez[12], Luis Amador[13], Raquel Betancourt[2], Salomón M. Ramírez-Jaramillo[5], Bruno Timbe-Borja[9], Miguel Gómez Laporta[5], Juan Fernando Webster Bernal[14], Luis Alfredo Oyagata Cachimuel[14], Daniel Chávez Jácome[13], Valentina Posse[7], Carlos Valle-Piñuela[14], Daniel Padilla Jiménez[14], Juan Pablo Reyes-Puig[2,15], Andrea Terán-Valdez[16], Luis A. Coloma[16], Ma. Beatriz Pérez Lara[2], Sofía Carvajal-Endara[16], Miguel Urgilés[2], Mario H. Yánez Muñoz[2]

**1** Grupo de Biogeografía y Ecología Espacial, Universidad Regional Amazónica Ikiam, Tena, Ecuador, **2** Instituto Nacional de Biodiversidad, Casilla, Quito, Ecuador, **3** Instituto de Diversidad Biológica Tropical iBIOTROP, Museo de Zoología & Laboratorio de Zoología Terrestre, Universidad San Francisco de Quito USFQ, Quito, Ecuador, **4** Universidad de La Laguna, Santa Cruz de Tenerife, España, **5** Proyecto PARG, Ministerio del Ambiente y Agua, PNUD, Quito, Ecuador, **6** Grupo de Investigación Evolución y Ecología de Fauna Neotropical, Facultad de Ciencias, Universidad Nacional de Colombia, Bogotá D.C., Colombia, **7** Laboratorio de Ecología Tropical y Servicios Ecosistémicos (EcoSs-Lab), Departamento de Ciencias Biológicas, Universidad Técnica Particular de Loja, Loja, Ecuador, **8** Parque Nacional Cajas ETAPA EP, Cuenca, Ecuador, **9** Museo de Zoología de la Universidad del Azuay, Cuenca, Ecuador, **10** Instituto de Ecología, A. C., Xalapa, México, **11** Laboratorio de Biología Evolutiva, Colegio de Ciencias Biológicas y Ambientales COCIBA, Instituto Biósfera USFQ, Universidad San Francisco de Quito, Quito, Ecuador, **12** Centro de Conservación de Anfibios AMARU, Cuenca, Ecuador, **13** Instituto de Ciencias Ambientales y Evolutivas, Doctorado en Ciencias m. Ecología y Evolución, Universidad Austral de Chile, Valdivia, Chile, **14** Ministerio del Ambiente y Agua, Quito, Ecuador, **15** Fundación Ecominga/Fundación Oscar Efrén Reyes, Baños, Ecuador, **16** Centro Jambatu de Investigación y Conservación de Anfibios, Fundación Jambatu, Quito, Ecuador

\* mauricio.ortega@ikiam.edu.ec

**Data Availability Statement:** All relevant data are within the paper and its Supporting Information files.

## Abstract

Ecuador is one of the most biodiverse countries in the world, but faces severe pressures and threats to its natural ecosystems. Numerous species have declined and require to be objectively evaluated and quantified, as a step towards the development of conservation strategies. Herein, we present an updated National Red List Assessment for amphibian species of Ecuador, with one of the most detailed and complete coverages for any Ecuadorian taxonomic group to date. Based on standardized methodologies that integrate taxonomic work, spatial analyses, and ecological niche modeling, we assessed the extinction risk and identified the main threats for all Ecuadorian native amphibians (635 species), using the IUCN Red List Categories and Criteria. Our evaluation reveals that 57% (363 species) are categorized as Threatened, 12% (78 species) as Near Threatened, 4% (26 species) as Data Deficient, and 27% (168 species) as Least Concern. Our assessment almost doubles the number of threatened species in comparison with previous evaluations. In addition to

**Funding:** This work was supported by the following projects: "Conservation of Ecuadorian Amphibians and access to genetic resources-PARG" managed by the Ministry of Environment and Water of Ecuador. The funder Project PARG provided support in the form of salaries for authors [KGLVM, SMRJ, MGL, DCJ], but did not have any additional role in the study design, data collection and analysis, decision to publish, or preparation of the manuscript. The specific roles of these authors are articulated in the 'author contributions' section. Project "On the quest of the golden fleece in Amazonia: The first herpetological DNA - barcoding expedition to unexplored areas on the Napo watershed, Ecuador", funded by the Secretaría Nacional de Ciencia y Tecnología del Ecuador (Senescyt- ENSAMBLE Grant #PIC-17-BENS-001), The World Academy of Sciences (TWAS Grant #16-095) granted to HMOA; the project Critical Ecosystem Partnership Fund (CEPF) grant CEPF-108984 "Amphibian Conservation in the Abra de Zamora Key Biodiversity Area of Ecuador" granted to DAO, PS, and DS; "Respuestas a la crisis de biodiversidad: La descripción de especies como herramienta de conservación, INÉDITA PIC-20-INE-USFQ-001" funded by Senescyt and granted to JMG; "USFQ-HUBI ID 48 Taxonomía, Biogeografía y Conservación de Anfibios y Reptiles" and "USFQ-HUBI ID 1057 "Impact of habitat changes on the biological diversity of the northern tropical Andes" funded by grants by Universidad san Francisco de Quito, The International Union for Conservation of Nature IUCN and NatureServe (with the support from the National Science Foundation's Dimensions of Biodiversity program, award 1136586), and by Programa "Becas de Excelencia", Secretaría de Educación Superior, Ciencia, Tecnología e Innovación (SENESCYT), Ecuador granted to DFCH; Collaboration Grant 'Investigación para la conservación de especies de anfibios críticamente amenazadas HUBI 16871' and COCIBA Grant 'Investigación y conservación de las especies críticamente amenazadas de ranas arlequín (Bufonidae: Atelopus spp HUBI 16808) granted to JMG.

**Competing interests:** The authors have declared that no competing interests exist.

habitat loss, the expansion of the agricultural/cattle raising frontier and other anthropogenic threats (roads, human settlements, and mining/oil activities) amplify the incidence of other pressures as relevant predictors of ecological integrity. Potential synergic effects with climate change and emergent diseases (apparently responsible for the sudden declines), had particular importance amongst the threats sustained by Ecuadorian amphibians. Most threatened species are distributed in montane forests and paramo habitats of the Andes, with nearly 10% of them occurring outside the National System of Protected Areas of the Ecuadorian government. Based on our results, we recommend the following actions: (i) An increase of the National System of Protected Areas to include threatened species. (ii) Supporting the ex/in-situ conservation programs to protect species considered like Critically Endangered and Endangered. (iii) Focalizing research efforts towards the description of new species, as well as species currently categorized as Data Deficient (DD) that may turn out to be threatened. The implementation of the described actions is challenging, but urgent, given the current conservation crisis faced by amphibians.

## Introduction

One of the main aims of conservation biology is to assess, understand, and mitigate threats to biodiversity. The International Union for Conservation of Nature (IUCN) Red List of Threatened Species is a powerful tool that allows not only to estimate species extinction risks but also to prioritize conservation efforts [1]. Red List Assessments are widely used by experts on several groups of plants and animals worldwide, as it applies standardized methods to assess threats and extinction risk, based on relevant quantitative and qualitative criteria [1–3].

Amphibians are one of the most diverse vertebrate groups in the Neotropical region [4]. In addition to presenting an extraordinary richness specific to each ecosystem, they are one of the most threatened taxa [5]. Their ectothermy makes them particularly vulnerable to environmental changes, mainly related to temperature and humidity, but also to infectious diseases [6–8]. Habitat loss, climate change, and diseases represent important threats to their populations [6, 9–11].

Ecuador is one of the countries with the highest number of amphibian species [12–15]. Ecuadorian amphibians are considered among the most threatened in South America, due to increased rates of habitat loss and deforestation for cattle raising, agriculture, mining, and oil exploitation [16–21]. Some historically conspicuous genera (harlequin frogs [*Atelopus* spp.], marsupial frogs [*Gastrotheca* spp.], and Andean water frogs [*Telmatobius* spp.]) have suffered dramatic populations declines or extinctions [8, 22–24], that seem to be related to the fungal panzootic *Batrachochytrium dendrobatidis* [8], although other factors, such as climate change may also be related [10].

Based on data gathered in the IUCN Red List, amphibians are the most threatened vertebrates globally, and the proportion of threatened species increases more rapidly than birds and mammals [25–27]. By March 2021, from an estimated 8126 amphibian species, 7212 were evaluated (87% of the known species), and 2442 (34% of the evaluated species) were considered threatened [assessed as Critically Endangered (CR), Endangered (EN), or Vulnerable (VU)]. However, globally, the proportion of threatened amphibian species would increase in a range between 41 and 53% if we considered that several Data Deficient (DD) species are likely to be threatened by extinction [28, 29].

In 2004, the Global Amphibian Assessment (GAA) published by the IUCN, Conservation International and NatureServe categorized for the first time, the amphibians of Ecuador; subsequently updated in 2006 and 2008 (www.iucn-amphibians.org). As a result of this process, 447 amphibian species were evaluated, and 165 (37%) were found to be threatened or extinct [25]. In 2011, an updated assessment was published for Ecuadorian species [27], with 465 evaluated species, 142 (30.5%) of which were found to be threatened (CR, EN, or VU) and nearly 29% classified as DD.

Since 2015, the Ministry of Environment and Water (MAAE) of Ecuador has been leading the project "Conservation of Ecuadorian amphibian biodiversity and sustainable use of genetic resources". One of the main components of the project is focused on understanding the conservation status of the amphibians of Ecuador and update the national red list. The goals of our study are to a) evaluate and update the extinction risk status of Ecuadorian amphibians, b) analyze spatial patterns of threatened species related to endemism, protected areas, and ecological regions in Ecuador, and c) suggest actions towards an integrative methodology to evaluate species conservation status.

## Materials and methods

### Amphibian database compilation

In order to gather distribution data for Ecuadorian amphibians, we compiled occurrence records along the complete distributional range of each species from the databases of the following ecuadorian museums: Instituto Nacional de Biodiversidad (INABIO-DHMECN); Museo de Zoología, Universidad Técnica Particular de Loja (MUTPL); Museo de Zoología, Universidad del Azuay (MZUA); Museo de Zoología, Universidad Tecnológica Indoamérica (MZUTI); Museo de Zoología, Universidad San Francisco de Quito (ZSFQ); Centro Jambatu (CJ). We also compiled records from the following databases: Proyecto Conservación de Anfibios y Recursos Genéticos del Ministerio de Ambiente del Ecuador (MAE-PARG), databases: Global Biodiversity Information Facility (GBIF; https://www.gbif.org), iNaturalist (https://www.iNaturalist.org), VertNet (http://vertnet.org/), Batrachia (https://www.batrachia.com), SapoPediaEcuador (http://www.anfibiosecuador.ec/index.php?aw; [13]), Anfibios Ecuador—Bioweb, Museo de Zoología de la Pontificia Universidad Católica del Ecuador (QCAZ; https://bioweb.bio; [14]). In addition, we obtained unpublished data evidenced by voucher specimens, or photographs, collected in the field and shared by the authors during the Ecuadorian Red List Assessment workshops (S1 Table). The final dataset included data compiled up to 31th October 2020 (Fig 1).

We followed the nomenclature proposed by Grant *et al.* [30], Guayasamin *et al.* [31], Castroviejo-Fisher *et al.* [32], Hedges *et al.* [33] for Strabomantidae; all other taxa groups follow The Amphibian Species of the World [15]. Records from outside Ecuador were analyzed, error-checked, and improved with the same level of accuracy as the records from inside Ecuador [34], through a taxonomic assessment of specimens in scientific collections, validation of records based on biogeographic distribution, phylogenetics, taxonomic analyses, photographs published elsewhere [35–43], a systematic literature review, and by discussions with 33 expert herpetologist from all over the country, including the authors of this paper, held during eight workshops between 2017 and 2020. Workshop participants were distributed on boards according to taxonomic families and geographic regions. Red List authorities participated in all workshops, to guarantee the correct use and application of all IUCN categories and criteria at the national level (i.e., Diego F. Cisneros-Heredia, Amphibian Red List Authority Regional Coordinator for Ecuador; Stephanie Arellano, Programme Officer, IUCN Regional Office for South America).

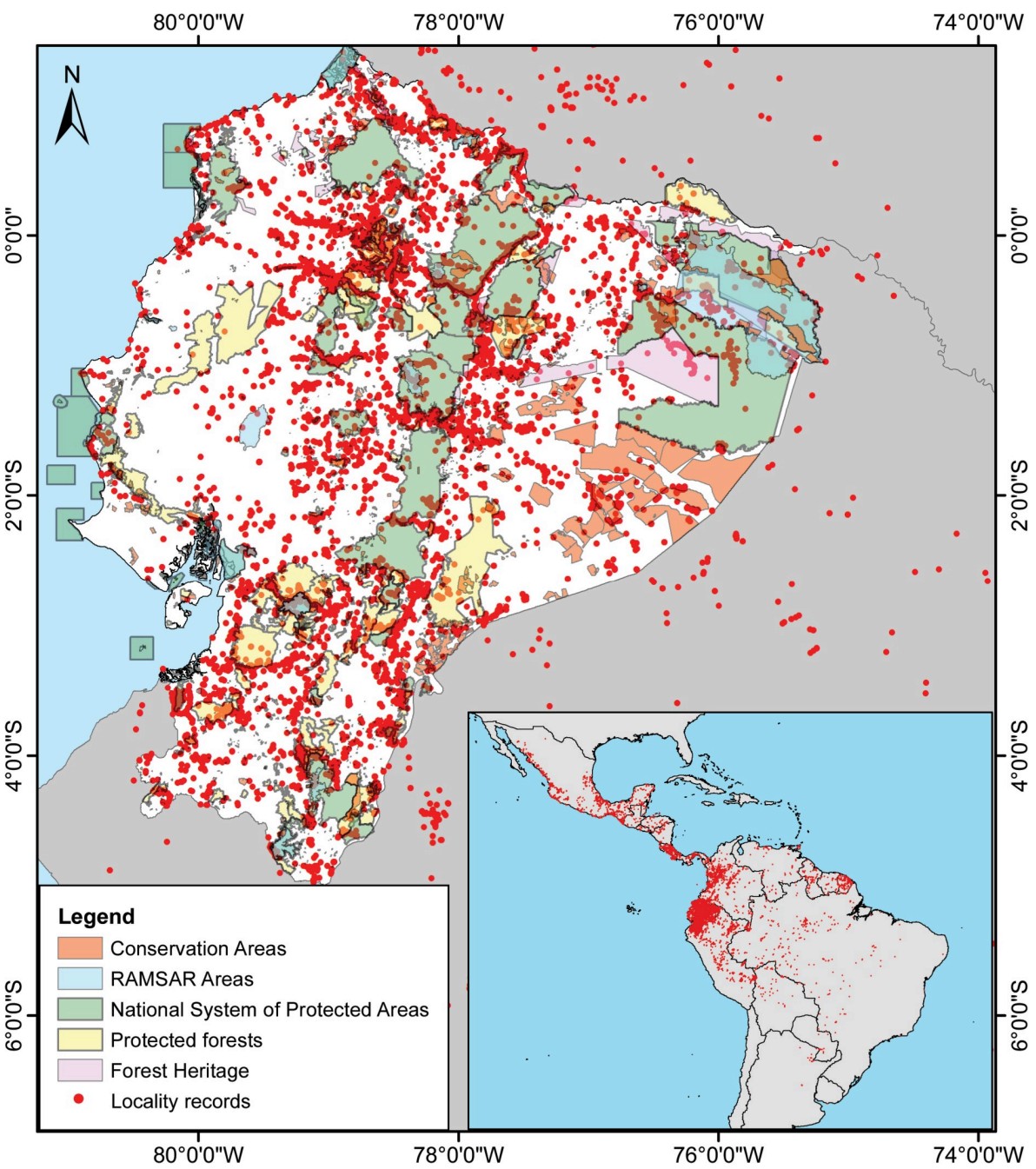

**Fig 1. Spatial distribution of records from the amphibian database.** A total of 37,328 records from 635 native species (including *Rana catesbeiana*, an introduced species) assessed for the IUCN Red List of Ecuadorian amphibians. Categories in the legend correspond to the National System of Protected Areas (SNAP, from Spanish acronym) in Ecuador. Details of collections, sources, and databases are provided in the S2 Table.

The Red List Assessment and the Red List of Amphibians of Ecuador herein presented were officially validated by the Ministry of Environment of Ecuador (Ministerial Agreement 069).

Spatialized data per species and geospatial data for watersheds, digital elevation model, and base maps from Ecuador (http://ide.ambiente.gob.ec/mapainteractivo/; http://www.igm.gob. ec/work/files/downloads/mapafisico.html) were revised using QGIS 3.4.14 to assess for data

consistency [44]. Elevation data was extracted for every data point and represented in boxplots to find outliers and other possible data errors. Problematic occurrence data, either at the georeferenced or taxonomic level, were removed from the dataset. Taxonomic experts validated the data and highlighted errors or inaccuracies during workshops. Records with incorrect georeferenced data were fixed using the Google Satellite layer in QGIS, only when the collectors verified the exact location. This process aimed to obtain a clean and debugged database that met appropriate standards for ecological niche modeling [45–47], biogeographic analyses [48, 49], and Red List Assessment [12, 16, 21], by Darwin Core guidelines (https://dwc.tdwg.org/).

## Environmental data

Climate variables for current and future scenarios were downloaded from the WorldClim2 database [50] (http://www.worldclim.org). We obtained 15 climatic variables at a 30 seconds (~1 km$^2$) spatial resolution; we excluded the four layers that combine precipitation and temperature information into the same layer due to spatial anomalies [51]. To characterize future climate conditions, we used data for two IPCC representative concentration pathways emissions scenarios (RCP 4.5 and 8.5) from the Hadley Global Environment Model 2—Earth System (HadGEM2-ES) global circulation model (GCM) [50]. Future RCP 4.5 scenarios assume relative slow income growth, increasing human population, and modest improvements in technology and energy intensity, leading to a higher demand for energy and increasing greenhouse gas emissions in the long-term considering an absence of climate change mitigation policies, whereas the RCP 8.5 scenario represents higher predicted greenhouse gas emissions [52].

## Data analysis and ecological modeling

Species were divided into two groups: 1) those that could be modeled (485 sp), and 2) those that could not be modeled due to the low number of occurrence points (fewer than 5 localities; 151 sp), occurrence points situated in closely-located pixels, or models not statistically significant by AUC thresholds. For the first group, we implemented a modeling process with Max-ENT [53]. We used the complete dataset of records along the known distributional range of each species. The characteristics of the model (creation, calibration, selection, and evaluation) were carried out in **kuenm** R package [54]. The Jackknife procedure and correlation statistics (-0.8 to 0.8 in *Pearson r* values) were used to assess the importance of the variables in a first run with all values by default.

Once the climate variables were selected, we obtained candidate models with different parameters (seven multiplication regulators—0.1, 0.4, 0.7, 1, 2, 3, 4—and seven feature classes —linear (*l*), quadratic (*q*), product (*p*), and all the combinations *lq*, *lp*, *pq*, *lqp*-). The maximum number of background points was 10,000. We randomly selected 70% of the data for training and used the remaining 30% for testing. A total of 500 runs were set for model building. The best model was selected according to the criteria of omission rate < 10% and delta AIC > 2.

An important step in ecological niche modeling is to define a calibration region, the accessible area ("M", hereinafter) for species [45, 47, 55]. In this study, we delimited "M" using the biogeographic provinces for the Neotropics [56], watersheds, and a digital elevation model to find the physical barriers that determine the accessibility area of each amphibian species. We found similar distribution patterns among several species, reiterating the same physical barriers (i.e. the Andes, basins, mountain ranges, etc.). For these reasons, some generic "M" were constructed for the different regions (i.e., highlands, coast, and Amazon), and these were assigned to each of the species. For taxa lacking enough data points for ecological modeling, the Area of Occupancy (AOO) was calculated [3] in R software (https://www.r-project.org/),

using a 2 x 2 km grid created in QGIS 3.4.14 and extracting and counting the number of cells occupied by the species.

## Cumulative species richness model

Cumulative species richness models (CSRM) were performed adding up the results of the Maxent binary models (suitability area) and Area of Occupancy (AOO) for each of the families and conservation categories. The results were shown using the ***tmap*** package [57] in R software (https://www.r-project.org/). Endemic species are herein referring to species restricted entirely to Ecuador, and were determined based on the categories proposed by Ron, A. (14). We used a Kruskal-Wallis test and a Wilcoxon test for paired samples to compare groups of endemic/ non-endemic taxa and conservation threat categories related to altitude ranges.

## Threat model

To have a better understanding of the potential impacts of human activities on the distribution of Ecuadorian amphibians, we followed standardized criteria to define risk elements and potential threats, based on expert supervision for hierarchical classification by IUCN-CMP (International Union for Conservation of Nature—Conservation Measures Partnership) [3, 58]. Overall, eight major threats with 34 subcategories were used to develop a threat or Environmental Risk Surface (ERS) model (Table 1).

We used a standard lexicon for classifications of threats [58]. These elements were spatially mapped (ArcMap v.10) as points, polygons, and lines, and then converted to raster files to calculate Euclidean distances of each threat. The Influence Distance (meters) was assigned to each subcategory based on buffer areas with a respective decay function, giving values according to the intensity of anthropogenic and natural threats. To reduce subjectivity by decision-making bias, regarding the ascription of Intensity to each risk element, we applied a Multi-Criteria Decision Making (MCDM) through Analytic Hierarchy Process (AHP) on the analysis (S3 Table). Once the inputs were obtained, the process was automatized using *ModelBuilder* from ArcMap, with an iterative process per subcategory (S1 Fig). Finally, the outputs were overlapped with a raster calculator to develop an ERS, which considers a weighted overlay of amphibians-specific threats in Ecuador, with a resolution of 30 m x 30 m.

## National Red List Assessment

The conservation status of amphibian species in Ecuador was assessed following the protocols, standards, criteria, subcriteria and adjustments for national assessments proposed by the IUCN [3, 59].

The dataset was compiled in a geospatial database used to assess the distribution and threats in a series of workshops promoted by the working group led by the authors. Data by species were analyzed mainly by records (N), percentage of records in Ecuador (%), area of occurrence (AOO, km$^2$), suitability area reconstructed by niche modeling (km$^2$), environmental contractions [60, 61] in future scenarios (% reduction relative to current ecological model), and values higher than 0.5 (in the third quartile) of the threat model.

All statistics (43 in total) used to apply criteria and subcriteria to assess the conservation status of a given species are detailed in S2 Table. Additional data related to population size or decline of the number of mature individuals were documented from literature or data from the authors provided in the workshops. As additional support for the evaluation, we used basic maps for National System of Protected Areas (SNAP—Sistema Nacional de Áreas Protegidas), Forestal Heritage, Protected forests and vegetation, Conservation Areas, Ramsar wetlands, Land Use and forested areas (until 2018) and Natural Regions of Ecuador, downloaded in

**Table 1. Major threats with their subcategories, influence distance, decay function, and Analytic Hierarchy Process (AHP) intensity value estimated for modeling threats to Ecuadorian amphibians.**

| Major threats and categories | Influence Distance (m) | Decay function | AHP Intensity |
|---|---|---|---|
| **Agriculture and aquaculture** | | | |
| Crops | | | |
| Permanent crops | 1875 | Logistic | 0.015 |
| Annual crops | 1250 | | 0.023 |
| Semi-permanent crops | 375 | | 0.016 |
| Grassland | 375 | | 0.036 |
| Agricultural mosaic | 375 | | 0.003 |
| Forest plantations | 250 | | 0.007 |
| Other agricultural lands | 125 | | 0.005 |
| Aquaculture | | | |
| Shrimp farm area | 1250 | MSSmall | 0.051 |
| **Biological resource use** | | | |
| Deforestation | 125 | Logistic | 0.34 |
| **Emerging diseases and Invasive species** | | | |
| Fungus *Chytridium* | 1250 | Constant | 0.035 |
| *Rana catesbeiana* (Bullfrog) | 1250 | Constant | 0.012 |
| **Energy production and mining** | | | |
| Operations | 1250 | MSSmall | 0.026 |
| Explorations | 1000 | | 0.005 |
| Mining and quarrying | | | |
| Concessions | 625 | | 0.002 |
| Construction Materials/Free use/Artisanal mining | 250 | | 0.007 |
| Oil drilling | | | |
| Active oil fields | 1250 | Logistic | 0.015 |
| Oil wells | 625 | | 0.021 |
| Dormant oil fields | 250 | | 0.004 |
| Oil blocks | 250 | | 0.003 |
| Hydroelectric power plants | | | |
| operative | 1250 | | 0.009 |
| Building | 625 | MSSmall | 0.012 |
| In project | 250 | | 0.001 |
| **Natural system modifications** | | | |
| Megaprojects area of influence | 1250 | MSSmall | 0.033 |
| **Population density** | | | |
| Population density | Continuous raster | Continuous | 0.22 |
| **Transportation** | | | |
| 1st order | 1250 | Lineal | 0.025 |
| 2nd order | 1000 | | 0.016 |
| 3rd order | 625 | | 0.011 |
| Roads | | | |
| Trails | 250 | | 0.008 |
| Airports | | | |
| Airports | 1250 | Logistic | 0.006 |
| Airport runways | 625 | | 0.003 |
| Oil pipeline/Polyduct | 625 | Logistic | 0.002 |
| Pipelines | | | |

*(Continued)*

**Table 1.** (Continued)

| Major threats and categories | Influence Distance (m) | Decay function | AHP Intensity |
|---|---|---|---|
| Gas pipeline | 250 | | 0.002 |
| **Stochastic events** | | | |
| Flood-prone areas | 625 | MSSmall | 0.005 |
| Volcanism area of influence | 12500 | MSSmall | 0.016 |

We used a standard lexicon for classifications of threats [58]. The Influence Distance (meters) was assigned to each subcategory based on buffer areas with a respective decay function, giving values according to the intensity of anthropogenic and natural threats. To reduce subjectivity by decision-making bias, regarding the ascription of Intensity to each risk element, we applied a Multi-Criteria Decision Making (MCDM) through Analytic Hierarchy Process (AHP) on the analysis (See S3 Table).

vector format from national servers [27, 62–64]. We calculated the threatened representativeness in a taxonomic group (TR): the number of threatened taxa / total number of taxa per family X 100. Comparative assessment of threatened taxa regarding the last National Red List follows Ron, Guayasamin (27).

## Results

### National Red List Assessment

A total of 126 databases belonging to various institutions and on-line resources were used to consolidate the dataset for the Ecuadorian amphibians (S1 Table). The final dataset included 37,328 records, of which 29,189 were located in Ecuador, of a total of 635 taxa (plus *Rana catesbeiana*, as an invasive species), which represent 100% of the species currently reported for Ecuador (Fig 2). GBIF, QCAZ, and INABIO were the data providers with the most representative collections included in the current Red List evaluation (Table 2).

Our IUCN Red List assessment resulted in the assignment of a threatened category (CR, EN, VU) to 57% of the Ecuadorian amphibian species, while 12% considered as Near Threatened (NT), 4% as DD, and 27% as Least Concern (LC) (Fig 2, Table 3). Eighty-five (85) taxa were considered as Critically Endangered CR (13.4%), including species of the genera *Atelopus* (24 spp.), *Hyloxalus* (9 spp.), and *Pristimantis* (12 spp.); 147 taxa (23.1%) were classified as Endangered (EN), and 131 (20.6%) as Vulnerable (VU). Strabomantidae is the family with the highest number of threatened taxa (CR = 18 spp, 3%, EN = 67 spp, 11.1%; VU = 87 spp, 14.5%, respectively). Strabomantidae (28.6%), Bufonidae (7%), and Centrolenidae (6.3%) harbor 42% of the total threatened species in Anura. An additional 78 taxa (12.3%) were evaluated as NT, and 168 as LC (26.4%). Finally, 26 taxa (4.1%) are considered as DD because the information was insufficient for an adequate assessment of their extinction risk (Fig 3, S3 Table). Regarding taxa under threatened categories, 56.7% (341 spp) of Anura, 72.7% (8 spp) of Caudata, and 60.9% (9 spp) of Gymnophiona qualified for one of these categories (Table 3). A total of 16 genera had all of their taxa considered as threatened [i.e., *Atelopus* (25 spp.), *Lynchius* (4 spp.), *Epicrionops* (3 spp.), *Telmatobius* (3 spp.), *Ctenophryne* (3 spp.), *Sachatamia* (3 spp.)]; seven genera had 70–90% of taxa as threatened [i.e., *Hyloxalus* (22 spp.), *Nymphargus* (15 spp.), *Gastrotheca* (14 spp.)]; 12 genera had 50–70% as threatened [i.e., *Pristimantis* (155 spp.), *Hyloscirtus* (13 spp.), *Caecilia* (7 spp.)] (Fig 3, S5 Table).

A total of 287 species (45%) of the species occurring in Ecuador are endemic. All endemic species of the families *Andinobates*, *Ectopoglossus*, *Paruwrobates*, and *Telmatobius* were considered to be threatened; the families Bufonidae, Dendrobatidae, Strabomantidae have 70–90% of their endemic species categorized as threatened. Eighteen genera have all of their endemic taxa evaluated as threatened (*i.e.*, *Atelopus*, *Lynchius*, *Niceforonia*, *Paruwrobates*,

(a) IUCN Red List Categories

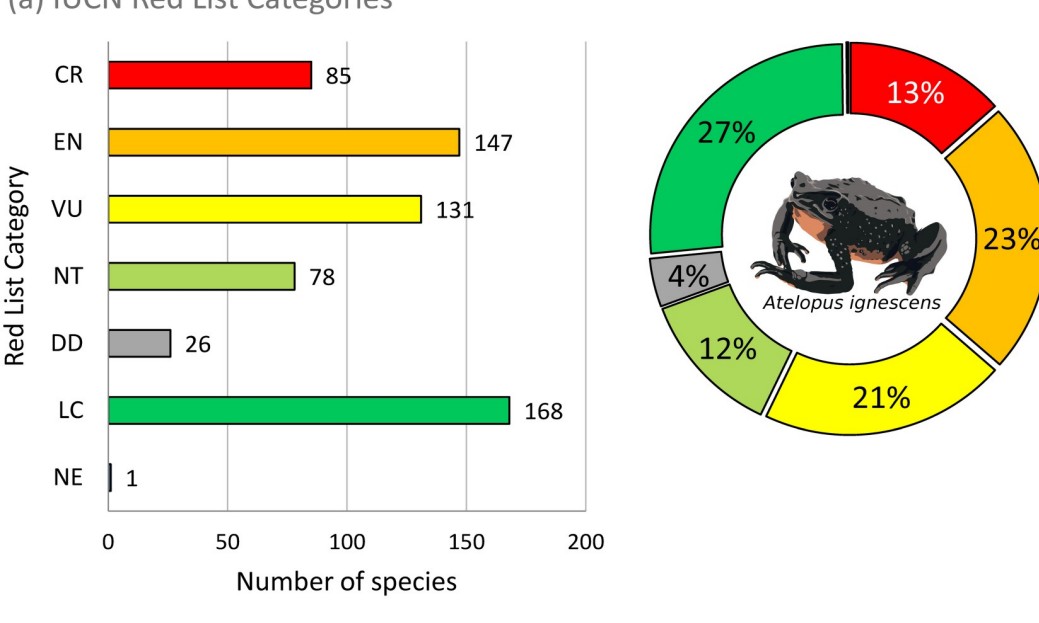

(b) Criteria

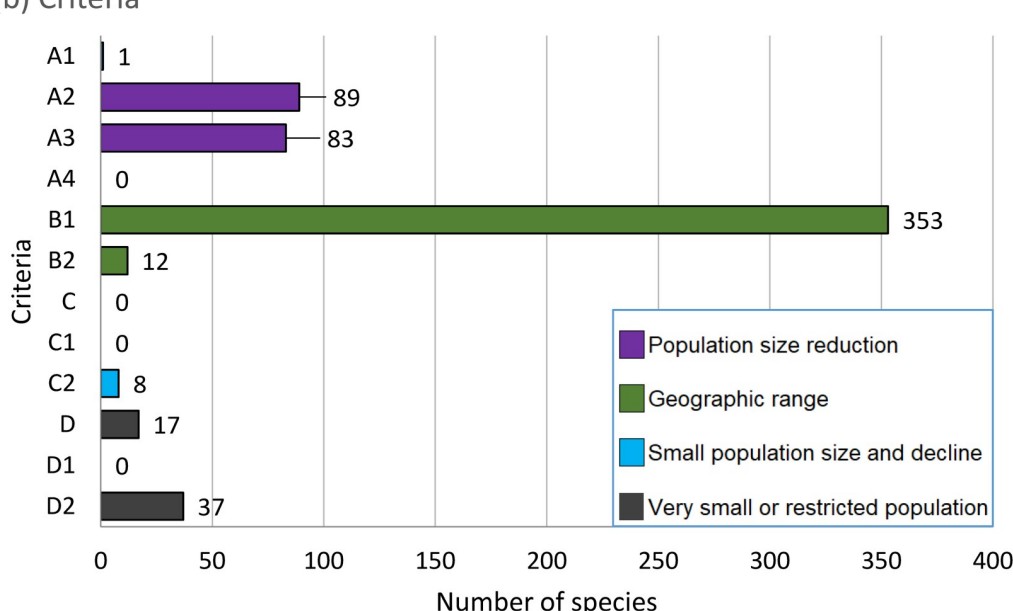

**Fig 2. IUCN Red List of amphibians from Ecuador.** The number of species by (a) Categories and (b) Criteria. Categories: CR = Critically Endangered, EN = Endangered, VU = Vulnerable, NT = Near Threatened, LC = Least Concern, DD = Data Deficient, NE = Not Evaluated—corresponds to *Rana catesbeiana*, an invasive species in Ecuador. *Atelopus ignescens* (Critically Endangered) was believed to be extinct until its rediscovery in 2016. Illustration by PARG.

*Rhaebo, Telmatobius)* and 10 genera have 70–90% of their endemic species as threatened (*Caecilia, Chiasmocleis, Epipedobates, Espadarana, Gastrotheca, Hyloscirtus, Hyloxalus, Nymphargus, Osornophryne, Pristimantis)* (S5 Table). Our assessment incorporates 172 species not previously evaluated; and we present the first extinction risk categorization for 127 species, previously classified as DD (Table 4, S3 Table).

**Table 2. Number of taxa and records analyzed from databases for the National Red List Assessment.**

| Collections Databases | CR | EN | VU | NT | LC | DD | NE* | Taxa (%) | Records (%) |
|---|---|---|---|---|---|---|---|---|---|
| Global Biodiversity Information Facility | 70 (702) | 107 (715) | 98 (1320) | 64 (1201) | 162 (11313) | 19 (122) | 1 (3) | 521 (82%) | 15376 (41%) |
| BIOWEB-PUCE | 47 (239) | 114 (1216) | 101 (1889) | 68 (2065) | 168 (10161) | 10 (211) | 1 (2) | 509 (80%) | 15783 (42%) |
| Instituto Nacional de Biodiversidad | 18 (29) | 81 (238) | 91 (314) | 59 (321) | 152 (1883) | 6 (31) | | 407 (64%) | 2816 (8%) |
| Museo de Zoología Universidad del Azuay | 4 (6) | 10 (56) | 28 (169) | 13 (43) | 84 (262) | | 1 (5) | 140 (22%) | 541 (1%) |
| Museo de Zoología, Universidad Técnica Particular de Loja | 8 (16) | 14 (71) | 31 (496) | 21 (320) | 57 (956) | | | 131 (21%) | 1859 (5%) |
| Red List Assessment Workshop | 8 (25) | 22 (58) | 13 (40) | 14 (29) | 40 (145) | 4 (9) | 1 (1) | 102 (16%) | 307 (1%) |
| Centro Jambatu | 8 (16) | 14 (20) | 17 (21) | 10 (51) | 32 (65) | () | | 81 (13%) | 173 (0%) |
| Escuela Politécnica Nacional | 5 (17) | 8 (12) | 11 (29) | 10 (19) | 19 (65) | 1 (1) | | 54 (8%) | 143 (0%) |
| Fundación Herpetológica Gustavo Orcés | 1 (1) | 3 (3) | 1 (1) | 7 (16) | 16 (64) | 1 (1) | | 29 (5%) | 86 (0%) |
| Batrachia | | 3 (30) | 4 (13) | 1 (1) | 2 (21) | | | 10 (2%) | 65 (0%) |
| Museo de Zoología Universidad Tecnológica Indoamérica | | | 2 (2) | 2 (2) | 4 (4) | 1 (1) | | 9 (1%) | 9 (0.02%) |
| Literature review | 2 (2) | | | | | | | 2 (0.3%) | 2 (0.01%) |
| Proyecto PARG | 2 (168) | | | | | | | 2 (0.3%) | 168 (0.5%) |
| **Total Species (records)** | **85 (636)** | **147 (2419)** | **131 (4294)** | **78 (4068)** | **26 (24939)** | **168 (376)** | **1 (11)** | | **37328** |

CR = Critically Endangered, EN = Endangered, VU = Vulnerable, NT = Near Threatened, LC = Least Concern, DD = Data Deficient, NE = No Evaluated, corresponds to *Rana catesbeiana*, an invasive species in Ecuador.

## Major threats

The ERS model is presented in S2 Fig. This model reveals high-risk areas (red) mainly located in the vicinity of large and medium-sized cities: Guayaquil (Coast), Quito (Andes), and Lago Agrio (Amazon). Medium-to-high-risk areas (orange) are primarily placed on the eastern and western foothills of the Andes mountain range, northern Amazonia, and northern Coast, with high threats scattered on central Coast and Amazonia regions, nearby roads. Medium-risk areas (yellow) can be identified along the Andes, as well as in the center-southern part of the Coast. We noticed that areas of low impact (green) are isolated, related to protected areas, inaccessible forests, and mountain ranges located in northwestern Ecuador, the Amazonian foothills of the Andes, and southern Amazonia (Fig 4).

Agriculture, transport, infrastructure (i.e., roads, oil pipelines, etc.), production areas (mining, oil camps, etc.), and deforestation are the most important threats for Ecuadorian amphibians, with 70–98% taxa associated with each of these categories (Fig 5, S7 Table). Near to 21–36% of assessed species will have a contraction in more than half of the area that represents their ecological niches (loss of environmental conditions, RCP 45/85) in future scenarios. We documented the presence of *Rana catesbeiana*, an introduced species, in several locations mainly distributed in the southern slopes of the Andes and coastal regions.

## Biogeographical patterns

Cumulative species richness models (CSRM) by threat category are shown in Fig 6 (models per species, genera, and families are detailed in Supplementary Material SM4). CSRM for threatened species generated a maximum value of 57 species overlapped per pixel. A high concentration of threatened taxa occur in the northern montane forests on both sides of the

**Table 3. Species (percentage) and threat categories, assessed by family in Ecuadorian amphibians.**

| Class/Families | CR | EN | VU | NT | LC | DD | Threatened Taxa | Total Taxa | TR (%) |
|---|---|---|---|---|---|---|---|---|---|
| **Anura** | **81 (13.5%)** | **136 (22.6%)** | **124 (20.6%)** | **78 (13%)** | **162 (27%)** | **20 (3.3%)** | **341 (56.7%)** | **601 (100%)** | **56.7%** |
| Aromobatidae | | 2 (0.3%) | | | 4 (0.7%) | 1 (0.2%) | 2 (0.3%) | 7 (1.2%) | 28.6% |
| Bufonidae | 29 (4.8%) | 7 (1.2%) | 6 (1%) | | 12 (2%) | 2 (0.3%) | 42 (7%) | 56 (9.3%) | 75% |
| Centrolenidae | 8 (1.3%) | 22 (3.7%) | 8 (1.3%) | 7 (1.2%) | 11 (1.8%) | 4 (0.7%) | 38 (6.3%) | 60 (10%) | 63.3% |
| Ceratophryidae | | | 1 (0.2%) | | 1 (0.2%) | 1 (0.2%) | 1 (0.2%) | 3 (0.5%) | 33.3% |
| Craugastoridae | | | | | 1 (0.2%) | | | 1 (0.2%) | 0% |
| Dendrobatidae | 10 (1.7%) | 12 (2%) | 9 (1.5%) | 9 (1.5%) | 7 (1.2%) | | 31 (5.2%) | 47 (7.8%) | 66% |
| Eleutherodactylidae | | 1 (0.2%) | | | 1 (0.2%) | | 1 (0.2%) | 2 (0.3%) | 50% |
| Hemiphractidae | 7 (1.2%) | 8 (1.3%) | 1 (0.2%) | 7 (1.2%) | 2 (0.3%) | | 16 (2.7%) | 25 (4.2%) | 64% |
| Hylidae | 5 (0.8%) | 14 (2.3%) | 6 (1%) | 18 (3%) | 55 (9.2%) | 2 (0.3%) | 25 (4.2%) | 100 (16.6%) | 25% |
| Leptodactylidae | 1 (0.2%) | 2 (0.3%) | 2 (0.3%) | 2 (0.3%) | 18 (3%) | | 5 (0.8%) | 25 (4.2%) | 20% |
| Microhylidae | | 3 (0.5%) | 2 (0.3%) | 1 (0.2%) | 5 (0.8%) | 1 (0.2%) | 5 (0.8%) | 12 (2%) | 41.7% |
| Pipidae | | | | | 1 (0.2%) | | | 1 (0.2%) | 0% |
| Ranidae | | | | 1 (0.2%) | 2 (0.3%) | | | 3 (0.5%) | 0% |
| Strabomantidae | 18 (3%) | 67 (11.1%) | 87 (14.5%) | 33 (5.5%) | 42 (7%) | 9 (1.5%) | 172 (28.6%) | 256 (42.6%) | 67.2% |
| Telmatobiidae | 3 (0.5%) | | | | | | 3 (0.5%) | 3 (0.5%) | 100% |
| **Caudata** | **3 (27.3%)** | **5 (45.5%)** | | | **2 (18.2%)** | **1 (9.1%)** | **8 (72.7%)** | **11 (100%)** | **72.7%** |
| Plethodontidae | 3 (27.3%) | 5 (45.5%) | | | 2 (18.2%) | 1 (9.1%) | 8 (72.7%) | 11 (100%) | 72.7% |
| **Gymnophiona** | **1 (4.3%)** | **6 (26.1%)** | **7 (30.4%)** | | **4 (17.4%)** | **5 (21.7%)** | **14 (60.9%)** | **23 (100%)** | **60.9%** |
| Caeciliidae | | 5 (21.7%) | 4 (17.4%) | | 3 (13%) | 4 (17.4%) | 9 (39.1%) | 16 (69.6%) | 56.3% |
| Rhinatrematidae | 1 (4.3%) | 1 (4.3%) | 1 (4.3%) | | | | 3 (13%) | 3 (13%) | 100% |
| Siphonopidae | | | 1 (4.3%) | | 1 (4.3%) | | 1 (4.3%) | 2 (8.7%) | 50% |
| Typhlonectidae | | | 1 (4.3%) | | | 1 | 1 (4.3%) | 2 (8.7%) | 50% |
| **Total general** | **85 (13.4%)** | **147 (23.1%)** | **131 (20.6%)** | **78 (12.3%)** | **168 (26.5%)** | **26 (4.1%)** | **363 (57.2%)** | **635 (100%)** | **57.2%** |

CR = Critically Endangered, EN = Endangered, VU = Vulnerable, NT = Near Threatened, LC = Least Concern, DD = Data Deficient. Pale red-shaded numbers are highlighted for families with the highest number of species in each threatened category. Threatened representativeness (TR): (number of threatened taxa / total number of taxa per family)*100.

Andes, paramos, and valleys in the central Andes and eastern montane forest along the Cutucú and Condor mountain ranges and foothills of the Amazon basin (Fig 6).

CSRM of CR taxa generated a maximum value of 12 species overlapped per pixel. A high concentration of taxa is located along both sides of the Andes, in northern Ecuador near the Cayambe Coca Ecological Reserve and Napo Sumaco-Galeras National Park, and the montane forest of southeastern Ecuador close to the Cutucú and Condor Mountain ranges. Models for EN taxa generated a maximum value of 28 species overlapped per pixel. The highest concentration of taxa was in the northwestern Andes, in areas west of the Pichincha volcano, Mindo, Guayllabamba basin in the provinces of Esmeraldas, Pichincha, Imbabura, and Carchi. Models for VU taxa generated a maximum value of 27 species overlapped per pixel. The higher concentration of VU taxa was located along with mountain forests and foothills in both sides of the Andes, in the Chocó region, in nearby areas of Napo Sumaco-Galeras National Park, and southeastern Ecuador (Fig 6).

Locality records of threatened species revealed differential patterns of distribution depending on the family (Fig 7). For example, threatened taxa of Bufonidae, Centrolenidae, Dendrobatidae, and Strabomantidae are related to the Andes and foothills. Telmatobiidae, which have all of their species categorized as CR, is restricted to the southern Andes (Fig 7). Strabomantidae is the only family that presents CR taxa limited to the coastal region. On the other hand,

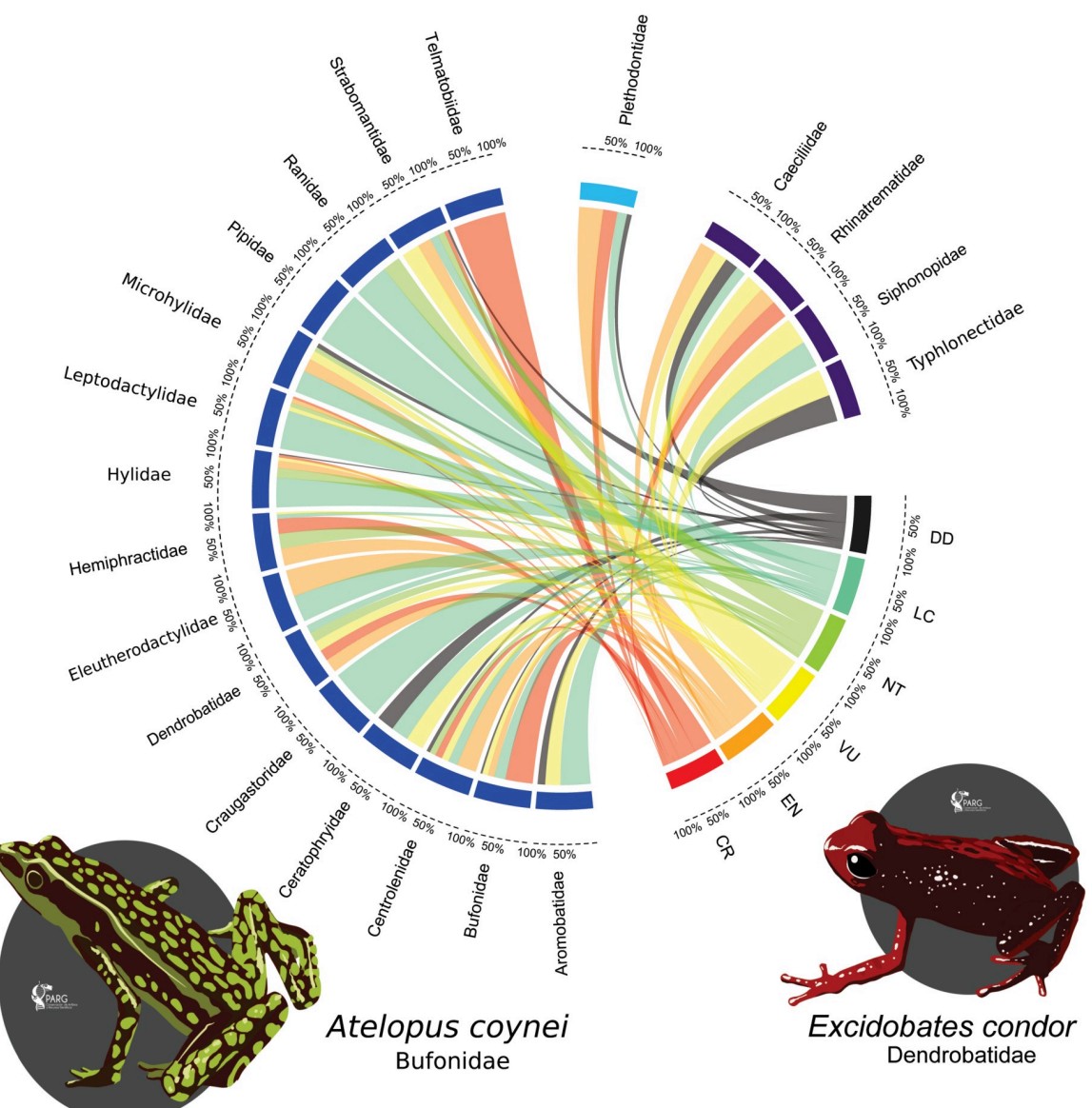

**Fig 3. A taxonomic perspective of the Red List status of amphibians in Ecuador.** The species composition (% of threatened species) of each family in Anura (dark blue), Caudata (bright blue), and Gymnophiona (purple) is characterized by ribbons connected to the current Red List status for each species. The numerical values below each family name depict the relative percentage with the associated Red List category: CR = Critically Endangered, EN = Endangered, VU = Vulnerable, NT = Near Threatened, LC = Least Concern, and DD = Data Deficient. Two endemic and threatened frogs are illustrated by *Atelopus coynei* (Critically endangered) distributed in northern Andes of Ecuador, whereas *Excidobates condor* (Endangered) is distributed in the Cordillera del Condor, southeastern Ecuador. Both species are threatened by habitat loss, mining, and climate change. Illustrations by PARG.

threatened taxa of Hylidae and Leptodactylidae have been recorded on both sides of the Andes, related to foothills and tropical forests. Threatened salamanders (Caudata, Plethodontidae) have been registered in northern Ecuador, towards foothills on both sides of the Andes, and tropical forests in the Chocó region (Fig 7).

Records of NT taxa are distributed on both sides of the Andes for Centrolenidae, Dendrobatidae, and Hylidae; while Hemiphractidae and Leptodactylidae are represented mainly in the Amazon basin and eastern slopes of the Andes. A wider distribution of locality records in

**Table 4. Number of amphibian species that changed their conservation status from the previous Ecuadorian Red List [27].**

| Previous assessment (2011) | Current assessment (2021) | | | | | | Total |
|---|---|---|---|---|---|---|---|
| | CR | EN | VU | NT | LC | DD | |
| CR | 29 | 9 | 1 | | 1 | 1 | 41 |
| EN | 10 | 28 | 20 | 3 | 3 | 1 | 65 |
| VU | 7 | 9 | 12 | 9 | 3 | | 40 |
| NT | 5 | 12 | 16 | 15 | 11 | | 59 |
| LC | 1 | 1 | 7 | 15 | 98 | 2 | 124 |
| DD | 17 | 41 | 23 | 14 | 32 | 8 | 135 |
| NE | 2 | 7 | 10 | 2 | 4 | 3 | 28 |
| Total | 71 | 107 | 89 | 58 | 152 | 15 | 492 |

Diagonal: Taxa that maintained the same conservation category between assessments. Upper diagonal: Taxa that decrease their conservation category; Lower diagonal: Taxa that increase their conservation category.

Ecuador (except the dry area in the coastal region) of NT taxa is identified for Strabomantidae. DD taxa are mostly located in the foothills and lowlands along the Amazon region, mainly for Bufonidae, Hylidae, Aromobatidae, and Centrolenidae; also, DD species in families Strabomontidae have been registered in the Andes (Fig 8).

The database had records from lowlands to highlands in Ecuador (min = 6 m, 1st Qu. = 821 m, median = 1694 m, mean = 1760 m, 3rd Qu. = 2728 m, max. = 5299 m). We report differences in the distribution of threatened and endemic taxa related with altitude [KW test ($\chi^2$) = 591.58, d.f. = 5, $p<2.2e$-16]. Threatened species were more frequently distributed on the highlands, montane areas, and foothills of the Andes i.e., CR (median = 2240 m, n = 1159), EN (median = 1862 m, $n$ = 2096), VU (median = 1533 m, $n$ = 3599), compared with NT taxa (Fig 7).

The highest number of threatened species was essentially encountered in three natural regions: eastern montane (318 spp), western montane (224 spp), and the Amazon (208 spp). Regarding species richness in each region, the paramo had the highest proportion of threatened species (80%), followed by the western montane (74%), Andean shrub (69%), and western foothills (65%). (Fig 9, S6 Table).

The "Vegetation and Protected Forests" category and the SNAP protected areas are the most important types of protected areas for threatened amphibian species, with an overall record of 203 (32%) and 196 species (31%), respectively (Fig 10, S6 Table). Sixty-five species (10%) do not occur in any protected area, with 26 rated as CR, 25 as EN, and 14 as VU.

## Discussion

### The current conservation status of Ecuadorian amphibians

The conservation status of 635 native amphibian species documented for Ecuador to date was assessed. The introduced species *R. catesbeiana* was considered to be invasive (S4 Table). Herein we report that 57% of the evaluated amphibian species are classified under some extinction risk using the IUCN Red List guidelines (13% CR, 23% EN, and 21% VU), with a further 12% falling into the NT category, and 4% listed as DD taxa. Our data present a rather pessimistic situation for one of the most diverse countries in amphibian species in the world [15]. This is especially true as the data are correlated with a high deforestation rate [20, 65], an immense pressure for mining development [66], and an important expected human population growth in the future [67].

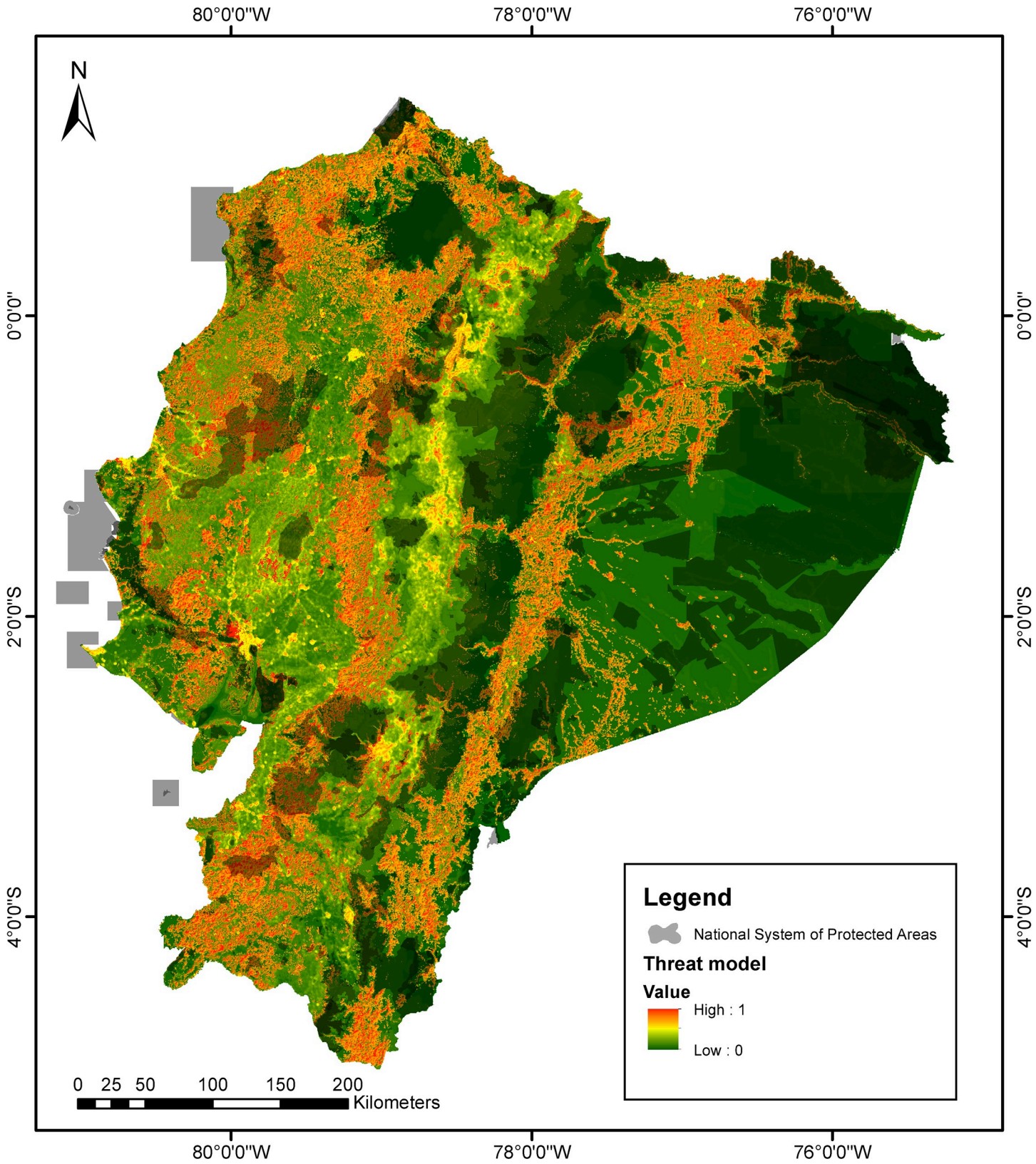

**Fig 4. High resolution (30 m x 30 m) Environmental Risk Surface (ERS) model for Ecuadorian amphibians.** Values of the ERS range from 0 (Green, low) to 1 (Red, high) to represent threat intensity. Shaded areas correspond to the National System of Protected areas shown in Fig 1.

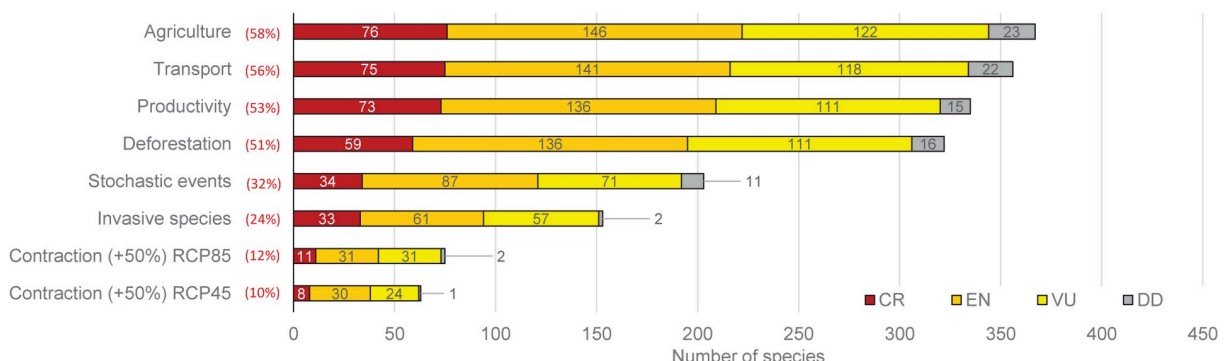

**Fig 5. Major threats associated with amphibian taxa (% of locality records in the database) by conservation categories in Ecuador.** Environmental contractions on climate change scenarios for RPC4.5 and RPC 8.5 are shown for those species with more than 50% of shift.

Compared to a previous National Amphibian Red List [27], we add assessments for 144 species and additionally, provide a status evaluation for 135 species that were considered DD at that time (Table 4; S3 Table). As a result of our study, 190 maintained the same conservation

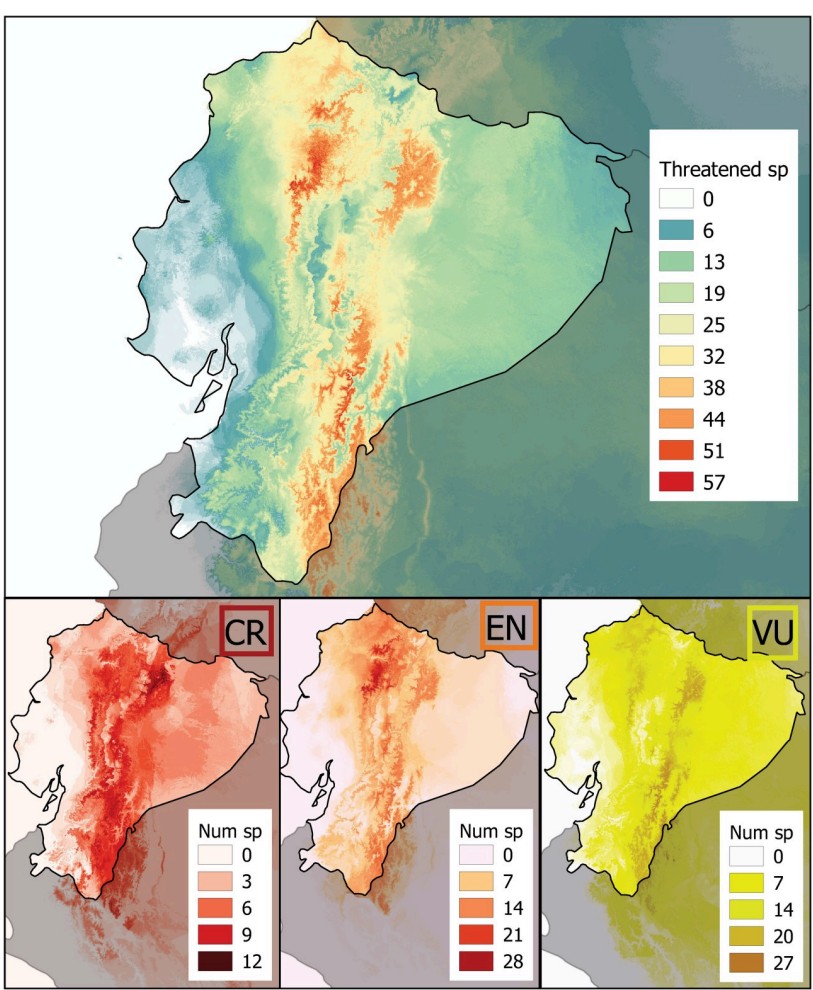

**Fig 6. Cumulative species richness for threatened taxa (*n* = 265 models) by Red List category.** Maps with cumulative species (Num sp) models per category and family are shown in S3–S5 Figs.

**Fig 7.** Occurrence data of threatened Ecuadorian amphibians by (a) taxonomic families, (b) endemic taxa to Ecuador, and (c) Red List categories in an altitudinal gradient. Risk categories: CR = Critically Endangered, EN = Endangered, VU = Vulnerable, NT = Near Threatened, DD = Data Deficient. Least Concern taxa have been removed.

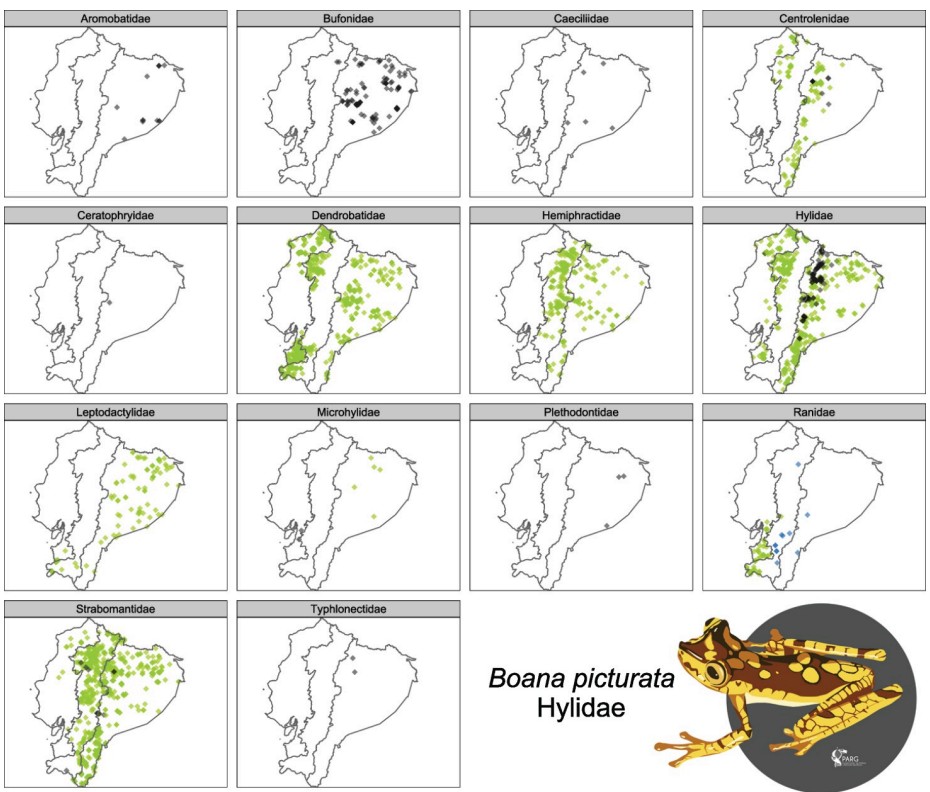

**Fig 8. Occurrence data of NT (green), DD (grey), and NE (blue) Ecuadorian amphibian species, by Red List category and family.** Only families with species in these categories are shown.

status, but 302 taxa have changed—168 species have now been found to qualify in a higher Red List category, while 64 have been assigned to a lower extinction risk category. The differences are explained by an increased knowledge for several groups, including taxonomic revisions with curated locality records, new information of recent species descriptions, ecological models for most of the threatened species, database compilation, and numerous evaluation workshops together with IUCN authorities and amphibian specialists [3, 27, 59].

Amidst a general trend of loss of biodiversity, some amphibian taxa (i.e. *Atelopus*, *Telmatobius*) show a phylogenetic vulnerability to anthropogenetic driven change and/or emerging diseases, most likely as a result of a combination of their distinctive life-history traits and immune constraints [23, 40, 68–71]. Because they contribute uniquely to the functioning of their communities, the loss of such species is especially worrisome as it is expected to have a disproportionate impact on the stability of local ecosystems, beyond their taxonomic loss [10]. This is of particular importance since most of them are endemic species not only to Ecuador, but also to specific habitats [14].

## Major threats

We have generated a quantitative and objective ranking of threats for Ecuadorian amphibians, using clear and comprehensive protocols [58]. A ranking of threats helps to identify and prioritize the conservation actions needed to mitigate them and provides results that are comparable and replicable [59]. Agriculture is of particular importance amongst the threats faced by Ecuadorian amphibians. In Ecuador, the unsustainable use of forested lands and agriculture/cattle-raising related deforestation, even in areas where the human population is low, are

## Biogeographic Regions

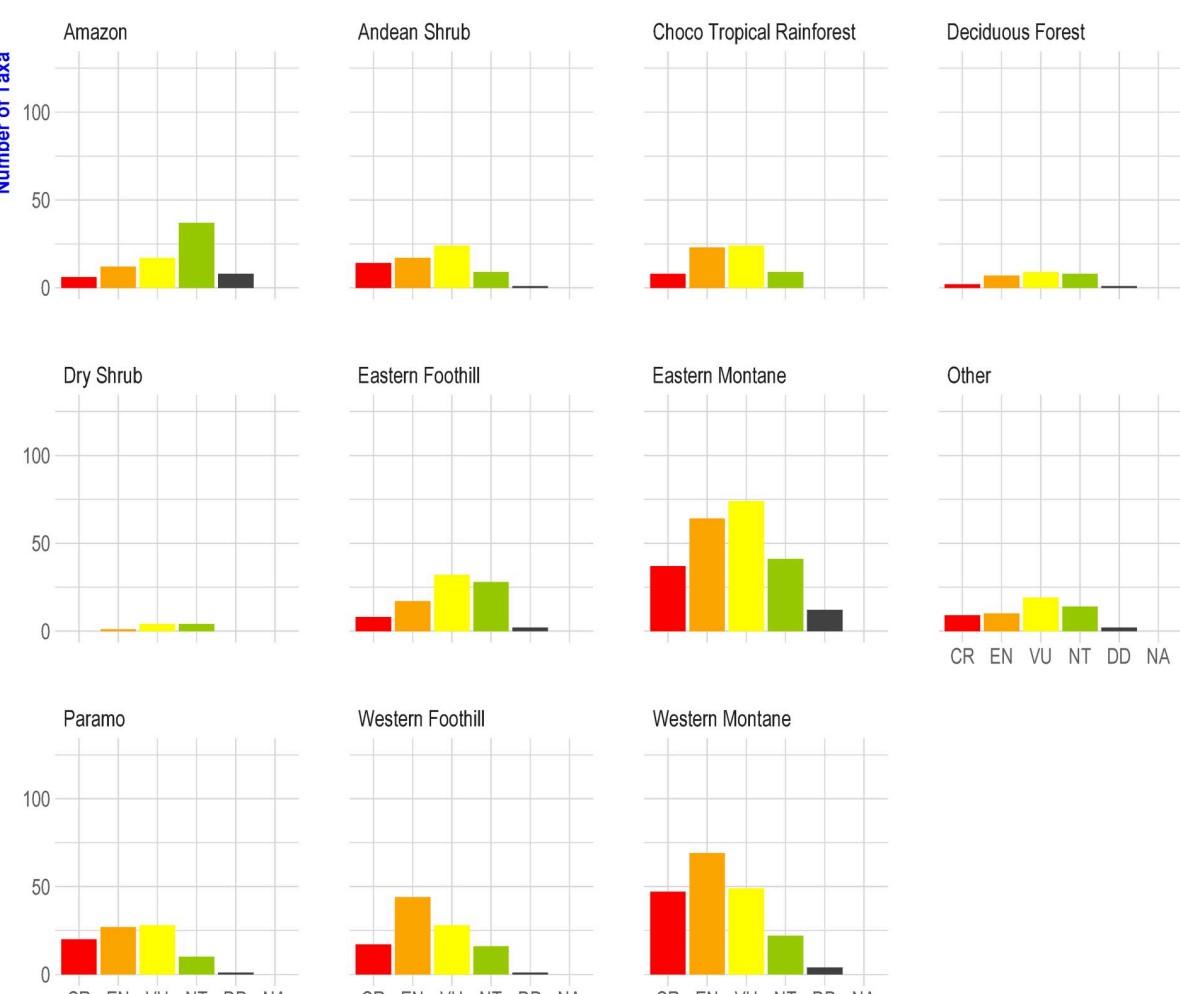

**Fig 9. Frequency of locality records of amphibians in each risk category by natural regions in Ecuador.** Categories: CR = Critically Endangered, EN = Endangered, VU = Vulnerable, NT = Near Threatened, DD = Data Deficient. Least Concern taxa have been removed from this figure.

conspicuous threats; therefore, a shift towards more sustainable and efficient agricultural practices is a priority. Also, some anthropogenic threats (roads, human settlements, and oil activities) amplify the incidence of other pressures and are the most relevant predictors of ecological integrity [1, 36, 72].

The ecological characteristics and microhabitat preferences of species can lead to deep variations in the susceptibility to certain drivers of extinction amongst taxa [2]. In amphibians, species respond differently to disturbance [73], therefore, conservation actions should take in consideration variables such as habitat specificity, life-history traits, distribution, connectivity, among others [74]. For example, we found a different distribution pattern in the case of threatened species, as well as endemic ones, both showing a higher density along an altitudinal gradient, with a peak in montane forests and highlands (Fig 7). However, cases of amphibian species interaction with spatial patterns of human impacts are puzzling. An alarming trend is that the greatest density of threatened taxa occurs in montane and paramo ecosystems, regions

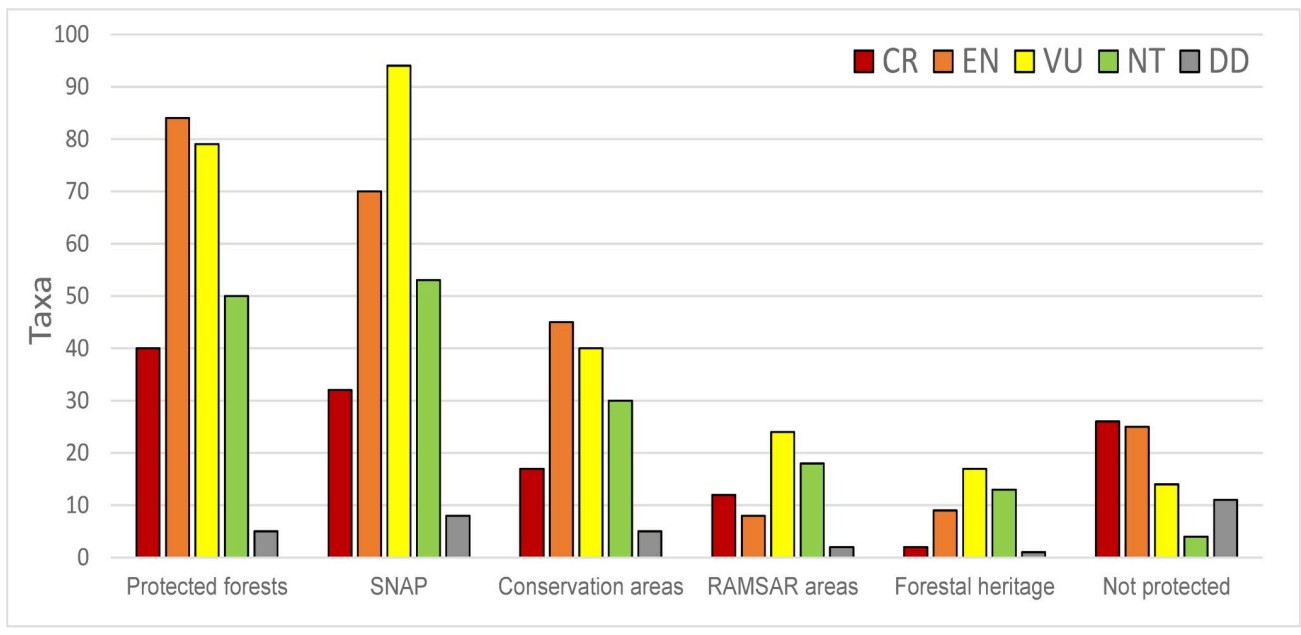

**Fig 10. The IUCN Red List of amphibians from Ecuador representation in the National System of Protected Areas.** Categories: CR = Critically Endangered, EN = Endangered, VU = Vulnerable, NT = Near Threatened, LC = Least Concern, DD = Data Deficient. SNAP–Governmental National System of Protected Areas, from the Spanish acronym.

that we would expect to be under a lesser anthropic impact. Further considerations on climate change and synergic effects with habitat loss and emergent diseases, like Chytridiomicosis, must be considered as major threats to Ecuadorian amphibians [10, 69], especially to endemic species.

Future assessment efforts should include the presence of invasive species as another potential threat to Ecuadorian amphibians. Currently, there are few studies focused on determining the expansion of these species and their effect on native amphibian populations. The bullfrog (*R. catesbeiana)* has been reported in six Ecuadorian provinces [75]; the rainbow trout (*Oncorhynchus mykiss*) and the common trout (*Salmo trutta*) are present in Andean areas of the whole Ecuadorian highlands [76]. The threat that these species represent to amphibians could be significant, considering their predatory and expansionist biology, but also because it generally overlaps with other threats that affect the habitat of species listed at extinction risk [77]. A special case of an invasive amphibian species (*Scinax quinquefasciatus*) potential negative effects on the environment can be found in the Galapagos Islands, reported in Santa Cruz and Isabela islands [14, 78, 79]. The effects of this species on the local ecosystems should be monitored in the future.

## Protected areas and threatened species

An evaluation of existing protected areas overlapping with the threatened species distribution reveals lacking of information from several areas in Ecuador, suggesting that much work is still needed to ensure the long-term survival of amphibians and their habitats. Since the existence of protected areas is considered the main hope for preserving threatened species from extinction [80], the fact that ~10% (65 species) of the Ecuadorian threatened amphibian species occur uniquely outside protected areas is alarming and highlights the limitations of the current National Protected Areas Network.

Our study emphasizes several areas that are home to a high number of threatened amphibian species and that are not protected (Fig 10). This is especially evident in three locations: the

Chocó area (north of the "Los Ilinizas" ecological reserve and the Pichincha volcano), the area among Cayambe-Coca, Antisana, and Sumaco, and in southern Ecuador (south of Sangay National Park). The national protected area network would maximize ecological representativeness and threatened species' coverage [81] if including areas with high amphibian species diversity (Figs 4, 6 and 10).

The dataset of distribution records reveals an important sampling effort bias, mostly related to roads or accessible areas (Fig 1). Large areas have been under-sampled, especially coastal areas, Andean paramo, and Amazonia. As a result, species categorized as DD are mostly located in the Amazon Region and on the eastern slopes of the Andes (Fig 8). In many cases, the remoteness of the areas prevents access due to logistical difficulties [81]. Although for the same reasons, the anthropic impact should be lower, in the case of high-altitude Andean habitats, we notice an overlapped high density of threatened species (especially CR), emphasizing the importance of focused searches for healthy populations in these secluded regions.

In the case of coastal areas, the shortage of inventories reflects insufficient sampling, severe habitat loss, but not necessary caused by limited access. Although a lower amphibian diversity is likely, mainly because of extreme climatic factors that restrict the distribution to a low number of resilient species, the total absence of records over large areas suggests a sampling bias [81]. However, the revision of threats indicates that the coastal region has a high proportion of its surface included under the highest risk, as well as a low representation in the Protected Areas Network (Fig 4). We emphasize the need for urgent base-line information regarding the amphibians inhabiting this region, as the lack of data makes it impossible to detect and monitor potential population declines or local extinctions.

## Towards an integrative methodology to evaluate the species conservation status

The methodology implemented herein is explicit, objective, and consistent, which are the main requisites to produce a solid assessment of species conservation categories. We are confident that we have produced standardized parameters to estimate robust risk variables that integrate interacting threats [2, 58] that are currently detailed in a methodological paper (López et al. in prep). We consider it as a key step in improving the protocol for Red List assessment in the effort to validate the taxonomic and spatial database. Ecological modeling was performed using all available data points along the known distributional range for nominal species, and as such included historical records, identifying and avoiding species complexes, and candidate new species based on phylogenetic evidence [36, 37, 40, 42, 82–85]. Although experts participated in the evaluation of the current status, the risk of extinction of species may be higher than assessed, due to the decline in their historical distribution range over time, as well as limitations on our understanding of population dynamics and ecological interactions [10, 86].

Demographic information is lacking for the vast majority of Ecuadorian amphibians, with 14% of the species assessed with limited data related to population size (Criteria A, C or D; Fig 2B). This constitutes a serious obstacle for obtaining a more comprehensive evaluation of their conservation status, preventing the early detection of declines. It is a particular case for Ecuador, where an important number of species are known only from a small number of specimens, and some have not been encountered for decades [e.g. 24]. This can be the result of cryptic habits that characterize some taxa (e.g. cecilians), but might as well indicate severe population declines or even extinctions (*Telmatobius*, *Atelopus*, or some centrolenids). This emphasizes the need for an intensive effort to gather base-line information on abundance and community composition for a diversity of amphibian populations.

Additionally, incomplete taxonomic delimitation has the potential to seriously impact amphibian conservation [71, 87]. Widely distributed species complexes, which are often assessed as LC, sometimes underlie cryptic taxa [37, 42, 84, 88], which might be facing particular conservation threats. We highlight the importance of taxonomy as a cornerstone for extinction risk assessments and conservation, especially in tropical mega-diverse regions. Assessments based on non-nominal species-level lineages or ambiguous names must be prioritized for taxonomic research [89].

## Conclusions

We offer the Red List Assessment of amphibian species in Ecuador, as one of the most detailed and complete taxonomic coverage for any Ecuadorian taxonomic group to date. Our evaluation of the 635 amphibian species recorded inside Ecuador found that 57% of all species are considered to be threatened (13% Critically Endangered, 23% Endangered, 21% Vulnerable). Moreover, 12% were listed as Near Threatened and 4% as Data Deficient. This assessment surprisingly almost doubled the number of species considered as threatened compared to the previous evaluation in 2011 [27]. Most threatened species are found in Andean montane forest and paramo, with nearly 10% of them located outside protected areas. To complement the results of this work and other future works, there is an urgent need for increasing the number of integrative taxonomic studies to describe new species and generate data on the ecology and genetics of populations and communities for those considered as taxonomic complexes. It is essential to focus research efforts towards species categorized as DD, that may be in danger of extinction [29, 90]. In parallel, new public and private reserves should target the protection of endangered species and their ecosystems. Conservation strategies should also strengthen *ex-situ* programs. Such integration will help in better management and conservation of amphibian species in hot-spot countries, like Ecuador.

## Supporting information

**S1 Table. Institution code, Institution name, database source, categories, number of records, and number of species assessed in the Red List for Ecuadorian amphibians.** (XLSX)

**S2 Table. Multi-Criteria Decision Making (MCDM) through the Analytic Hierarchy Process (AHP) for construct the threat model (Fig 4).** (XLSX)

**S3 Table. Species list of Ecuadorian amphibians, endemism, conservation areas, major threats, extinction risk criteria, subcriteria, and metrics used for the Red List Assessment.** (XLSX)

**S4 Table. The Red List for Ecuadorian amphibians, with details of criteria and subcriteria used for the evaluation of national categories.** Categories: CR = Critically Endangered, EN = Endangered, VU = Vulnerable, NT = Near Threatened, LC = Least Concern, DD = Data Deficient, NE = Not Evaluated, correspond to *Rana catesbeiana*, an invasive species in Ecuador. (XLSX)

**S5 Table. The number of taxa assessed by genera in the current evaluation, categories, and threatened representativeness in the group (%).** (XLSX)

**S6 Table. Species (percentage) and categories of threat assessed by type of protected area in Ecuador.** CR = Critically Endangered, EN = Endangered, VU = Vulnerable, NT = Near Threatened, LC = Least Concern, DD = Data Deficient. SNAP = National System of Protected Areas, from the Spanish acronym.
(XLSX)

**S7 Table. Species (percentage) and categories of the conservation status of amphibians by major threats in Ecuador.** CR = Critically Endangered, EN = Endangered, VU = Vulnerable, NT = Near Threatened, LC = Least Concern, DD = Data Deficient.
(XLSX)

**S8 Table. Species and categories of threat assessed by natural regions and Protected Area in Ecuadorian amphibians.** CR = Critically Endangered, EN = Endangered, VU = Vulnerable, NT = Near Threatened, LC = Least Concern, DD = Data Deficient.
(XLSX)

**S9 Table. Species (percentage) and categories of threat assessed by natural regions and provinces in Ecuadorian amphibians.** CR = Critically Endangered, EN = Endangered, VU = Vulnerable, NT = Near Threatened, LC = Least Concern, DD = Data Deficient.
(XLSX)

**S1 Fig. Automated procedure was designed using the *ModelBuilder* tool in ArcMap v.10 to perform threat model iterative analysis.**
(DOCX)

**S2 Fig. The threat model for Ecuadorian amphibians, raster image.** https://drive.google.com/file/d/1wkdx8DgDwKhVEyElDhc23wmEiknFw4DE/view?usp=sharing.
(DOCX)

**S3 Fig. Cumulative richness models of taxa qualified as critically endangered by family.**
(DOCX)

**S4 Fig. Cumulative richness models of taxa qualified as Endangered by family.**
(DOCX)

**S5 Fig. Cumulative richness models of taxa qualified as Vulnerable by family.**
(DOCX)

## Acknowledgments

We thank Pablo Larco, Patricia Pachacama, and Paola Guijarro (PARG Project), for their valuable support along with the red list assessment; to Andrea Coloma, Eduardo Toral, Stephanie Arellano, Ernesto Arbeláez, Diego Inclán, and Grace C. Reyes Ortega, for sharing information and technical comments along with the workshops and database curation.

## Author Contributions

**Conceptualization:** H. Mauricio Ortega-Andrade, Marina Rodes Blanco, Diego F. Cisneros-Heredia, Nereida Guerra Arévalo, Karima Gabriela López de Vargas-Machuca, Carolina Reyes-Puig, Paul Székely, Octavio R. Rojas Soto, Juan M. Guayasamin, Mario H. Yánez Muñoz.

**Data curation:** H. Mauricio Ortega-Andrade, Marina Rodes Blanco, Diego F. Cisneros-Heredia, Nereida Guerra Arévalo, Karima Gabriela López de Vargas-Machuca, Juan C. Sánchez-

Nivicela, Diego Armijos-Ojeda, José Francisco Cáceres Andrade, Carolina Reyes-Puig, Amanda Belén Quezada Riera, Paul Székely, Diana Székely, Juan M. Guayasamin, Fausto Rodrigo Siavichay Pesántez, Luis Amador, Raquel Betancourt, Salomón M. Ramírez-Jaramillo, Bruno Timbe-Borja, Miguel Gómez Laporta, Juan Fernando Webster Bernal, Luis Alfredo Oyagata Cachimuel, Daniel Chávez Jácome, Valentina Posse, Carlos Valle-Piñuela, Daniel Padilla Jiménez, Juan Pablo Reyes-Puig, Andrea Terán-Valdez, Luis A. Coloma, Ma. Beatriz Pérez Lara, Sofía Carvajal-Endara, Miguel Urgilés, Mario H. Yánez Muñoz.

**Formal analysis:** H. Mauricio Ortega-Andrade, Marina Rodes Blanco, Nereida Guerra Arévalo, Karima Gabriela López de Vargas-Machuca, Diego Armijos-Ojeda, Carolina Reyes-Puig, Octavio R. Rojas Soto, Diana Székely, Juan M. Guayasamin, Fausto Rodrigo Siavichay Pesántez, Raquel Betancourt, Bruno Timbe-Borja, Miguel Gómez Laporta, Daniel Chávez Jácome, Valentina Posse, Juan Pablo Reyes-Puig, Luis A. Coloma, Mario H. Yánez Muñoz.

**Funding acquisition:** H. Mauricio Ortega-Andrade.

**Investigation:** H. Mauricio Ortega-Andrade, Marina Rodes Blanco, Diego F. Cisneros-Heredia, Nereida Guerra Arévalo, Karima Gabriela López de Vargas-Machuca, Juan C. Sánchez-Nivicela, Diego Armijos-Ojeda, Carolina Reyes-Puig, Amanda Belén Quezada Riera, Paul Székely, Octavio R. Rojas Soto, Diana Székely, Juan M. Guayasamin, Fausto Rodrigo Siavichay Pesántez, Raquel Betancourt, Salomón M. Ramírez-Jaramillo, Bruno Timbe-Borja, Miguel Gómez Laporta, Juan Fernando Webster Bernal, Luis Alfredo Oyagata Cachimuel, Daniel Chávez Jácome, Valentina Posse, Daniel Padilla Jiménez, Juan Pablo Reyes-Puig, Luis A. Coloma, Ma. Beatriz Pérez Lara, Miguel Urgilés, Mario H. Yánez Muñoz.

**Methodology:** H. Mauricio Ortega-Andrade, Marina Rodes Blanco, Diego F. Cisneros-Heredia, Nereida Guerra Arévalo, Karima Gabriela López de Vargas-Machuca, Juan C. Sánchez-Nivicela, Diego Armijos-Ojeda, Carolina Reyes-Puig, Paul Székely, Octavio R. Rojas Soto, Diana Székely, Juan M. Guayasamin, Fausto Rodrigo Siavichay Pesántez, Raquel Betancourt, Salomón M. Ramírez-Jaramillo, Miguel Gómez Laporta, Daniel Padilla Jiménez, Juan Pablo Reyes-Puig, Luis A. Coloma, Ma. Beatriz Pérez Lara, Mario H. Yánez Muñoz.

**Project administration:** H. Mauricio Ortega-Andrade.

**Software:** Nereida Guerra Arévalo, Karima Gabriela López de Vargas-Machuca.

**Supervision:** H. Mauricio Ortega-Andrade, Marina Rodes Blanco, Juan C. Sánchez-Nivicela, Carolina Reyes-Puig, Paul Székely, Diana Székely, Juan M. Guayasamin, Fausto Rodrigo Siavichay Pesántez, Luis Amador, Raquel Betancourt, Bruno Timbe-Borja, Miguel Gómez Laporta, Mario H. Yánez Muñoz.

**Validation:** H. Mauricio Ortega-Andrade, Marina Rodes Blanco, Diego F. Cisneros-Heredia, Nereida Guerra Arévalo, Juan C. Sánchez-Nivicela, Diego Armijos-Ojeda, José Francisco Cáceres Andrade, Paul Székely, Diana Székely, Juan M. Guayasamin, Fausto Rodrigo Siavichay Pesántez, Raquel Betancourt, Salomón M. Ramírez-Jaramillo, Bruno Timbe-Borja, Miguel Gómez Laporta, Juan Fernando Webster Bernal, Luis Alfredo Oyagata Cachimuel, Daniel Chávez Jácome, Daniel Padilla Jiménez, Juan Pablo Reyes-Puig, Andrea Terán-Valdez, Luis A. Coloma, Ma. Beatriz Pérez Lara, Sofía Carvajal-Endara, Mario H. Yánez Muñoz.

**Visualization:** H. Mauricio Ortega-Andrade, Marina Rodes Blanco, Diego F. Cisneros-Heredia, Nereida Guerra Arévalo, Amanda Belén Quezada Riera, Paul Székely, Raquel Betancourt, Miguel Gómez Laporta, Valentina Posse, Juan Pablo Reyes-Puig, Sofía Carvajal-Endara, Mario H. Yánez Muñoz.

**Writing – original draft:** H. Mauricio Ortega-Andrade, Marina Rodes Blanco, Diego F. Cisneros-Heredia, Nereida Guerra Arévalo, Karima Gabriela López de Vargas-Machuca, Carolina Reyes-Puig, Amanda Belén Quezada Riera, Paul Székely, Octavio R. Rojas Soto, Diana Székely, Juan M. Guayasamin.

**Writing – review & editing:** H. Mauricio Ortega-Andrade, Marina Rodes Blanco, Diego F. Cisneros-Heredia, Nereida Guerra Arévalo, Juan C. Sánchez-Nivicela, Diego Armijos-Ojeda, José Francisco Cáceres Andrade, Carolina Reyes-Puig, Paul Székely, Octavio R. Rojas Soto, Diana Székely, Juan M. Guayasamin, Fausto Rodrigo Siavichay Pesántez, Luis Amador, Raquel Betancourt, Salomón M. Ramírez-Jaramillo, Bruno Timbe-Borja, Miguel Gómez Laporta, Juan Fernando Webster Bernal, Luis Alfredo Oyagata Cachimuel, Valentina Posse, Carlos Valle-Piñuela, Daniel Padilla Jiménez, Juan Pablo Reyes-Puig, Andrea Terán-Valdez, Luis A. Coloma, Ma. Beatriz Pérez Lara, Sofía Carvajal-Endara, Miguel Urgilés, Mario H. Yánez Muñoz.

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
