## [Decision Letter · Decision Letter 0]

25 Feb 2021

PONE-D-20-34320

Red List assessment for amphibian species of Ecuador: a multidimensional approach for their conservation

PLOS ONE

Dear Dr. Ortega-Andrade,

Thank you for submitting your manuscript to PLOS ONE. After careful consideration, we feel that it has merit but does not fully meet PLOS ONE’s publication criteria as it currently stands. Therefore, we invite you to submit a revised version of the manuscript that addresses the points raised during the review process.

The one non-anonymous referee and myself like the paper and find it makes an important contribution to EC amphibian conservation. The referee suggests

- to strengthen if global or regional/bational assessment,

- to better work out the differences between this and earlier assessments,

- that language might need some improvement,

apart from few others aspects (for detials see reviewer comments).

I agree that these things are relatively easy to manage and much improve the paper, which is very good anyway.

We look forward to receiving your revised manuscript.

Kind regards,

Stefan Lötters

Academic Editor

PLOS ONE

Journal Requirements:

"The authors have declared that no competing interest exist"

We note that one or more of the authors are employed by a commercial company: Independent researcher.

3.1. Please provide an amended Funding Statement declaring this commercial affiliation, as well as a statement regarding the Role of Funders in your study. If the funding organization did not play a role in the study design, data collection and analysis, decision to publish, or preparation of the manuscript and only provided financial support in the form of authors' salaries and/or research materials, please review your statements relating to the author contributions, and ensure you have specifically and accurately indicated the role(s) that these authors had in your study. You can update author roles in the Author Contributions section of the online submission form.

3.2. Please also provide an updated Competing Interests Statement declaring this commercial affiliation along with any other relevant declarations relating to employment, consultancy, patents, products in development, or marketed products, etc.  

4. We note that Figures 1, 4, 6, 7, 8 and S3, S4, S5 in your submission contain map images which may be copyrighted. All PLOS content is published under the Creative Commons Attribution License (CC BY 4.0), which means that the manuscript, images, and Supporting Information files will be freely available online, and any third party is permitted to access, download, copy, distribute, and use these materials in any way, even commercially, with proper attribution. For these reasons, we cannot publish previously copyrighted maps or satellite images created using proprietary data, such as Google software (Google Maps, Street View, and Earth). For more information, see our copyright guidelines: http://journals.plos.org/plosone/s/licenses-and-copyright.

4.1.    You may seek permission from the original copyright holder of Figures 1, 4, 6, 7, 8 and S3, S4, S5 to publish the content specifically under the CC BY 4.0 license. 

4.2.    If you are unable to obtain permission from the original copyright holder to publish these figures under the CC BY 4.0 license or if the copyright holder’s requirements are incompatible with the CC BY 4.0 license, please either i) remove the figure or ii) supply a replacement figure that complies with the CC BY 4.0 license. Please check copyright information on all replacement figures and update the figure caption with source information. If applicable, please specify in the figure caption text when a figure is similar but not identical to the original image and is therefore for illustrative purposes only.

Reviewers' comments:

Reviewer's Responses to Questions

**Comments to the Author**

1. Is the manuscript technically sound, and do the data support the conclusions?

Reviewer #1: Yes

2. Has the statistical analysis been performed appropriately and rigorously? 

Reviewer #1: Yes

3. Have the authors made all data underlying the findings in their manuscript fully available?

Reviewer #1: Yes

4. Is the manuscript presented in an intelligible fashion and written in standard English?

Reviewer #1: Yes

5. Review Comments to the Author

Reviewer #1: Manuscript: Red List assessment for amphibian species of Ecuador: a multidimensional approach for their conservation

Reviewer: Luis Fernando Marin da Fonte (IUCN Amphibian Specialist Group for Brazil, Amphibian Survival Alliance)

In this paper, the authors provide a very detailed and complete assessment of the conservation status of amphibians in Ecuador besides providing the most up to date list of species for the country. It is a very impressive and important work with great potential to benefit the conservation of this group of animals. Besides having a local importance, with the potential to inform public policies, it can also serve as a role model to other countries globally. I congratulate the authors for their work and recommend its publication with minor changes. That said, I have some suggestions to improve the overall quality of the manuscript and think that, before publication, some important issues should be addressed, as following:

1) with regard to the writing, I acknowledge that the authors are not native English speakers and admire their willingness to publish in a foreign language. Since I am also not a native English speaker, I cannot comment much on the language (though I have made some minor suggestions). The introduction reads very well, but the discussion reads a bit difficult sometimes, with too long and often confusing sentences. I suggest using shorter sentences to improve clarity.

2) my main criticism to this paper (though relatively easy to fix) is that it is not clear in the manuscript if this is a global or a regional/national assessment. Moreover, it is a bit confusing what are the ties of this assessment with the IUCN.

Global assessments (published on the IUCN Red List website) take into account the complete geographic range of taxa, while regional assessments take into account occurrence inside a specific region (usually inside country boundaries). Of course, for species endemic to a given country, global assessments will match national assessments, but this is not the case for species shared by different countries. The IUCN provides specific guidelines for regional assessments, but it is not clear if the authors followed them.

All things considered, one would imagine this is indeed a national assessment. However, things get a bit confusing when authors say that they have used the ‘complete distributional range’ of species (line 181). This would not be a problem per se, but it is not clear in the manuscript if the records from other countries were validated by experts - in the same way that records from inside Ecuador were validated by Ecuadorian experts in the workshops. Moreover, it is not clear if the search for records outside Ecuador was exhaustive or not (if I got it right, they were gathered only from online databases?). From what I have understood, records from outside Ecuador were not collected and analysed with the same level of accuracy than the records from inside Ecuador. If this is the case, you would have two kind of data: high quality data from inside Ecuador, and possibly incomplete/unreviewed/not validated data from outside Ecuador. If this is a regional/national assessment, this should not be a problem, since the most relevant data for the evaluations are the ones from inside Ecuador. But if this was intended to be a global assessment, then we would have a problem. In this sense, I think it is fundamental to make it clear, both in the title and in the body of the text, that (or if) this is a national assessment. Moreover, it is important to make clear if data from outside Ecuador was validated by experts and if the search for records was exhaustive (in other words, is this data comparable with the data from inside Ecuador?). Finally, it is important to show how exactly the data from outside Ecuador was used, since this is fundamental to determine if this is a global or a regional/national assessment.

Still in this topic, it is not clear if the authors followed the IUCN guidelines or only the IUCN criteria and categories. This might seem perfumery, but it is not. The IUCN guidelines inform how the criteria and categories should be used, and this is even more relevant in regional assessments. The criteria, in their turn, inform if and how the taxa fall under the different threat categories. In this sense, since the criteria and categories are the same for global and regional assessments, but the guidelines are not, it is not informative enough to only say that the assessment followed “IUCN criteria” (lines 68, 298), “IUCN standards” (lines 298, 459) or “IUCN protocols” (line 298), whatever ‘standard’ and ‘protocol’ mean in this context. It is important to know if the assessment followed the IUCN guidelines and, even more important, if it followed the guidelines for regional assessments.

Finally, it is important to make clear if this is an official IUCN assessment (i.e. if members of the IUCN Amphibian Red List Authority were present in the workshops and if the assessment was validated by external members of the IUCN Red List Authority). I specifically mention this because in line 330 the authors say “… the IUCN Red List assessment resulted in…”. If this is not an official IUCN assessment, you cannot say that. In this case, I suggest changing for “... the assessment following the IUCN Red List guidelines and criteria resulted...”. Same thing in line 151.

I also suggest reviewing the references to all IUCN documents. They are generally incomplete in the list of references and/or cited without criteria in the body of the text. For instance, the authors do cite the IUCN guidelines for regional assessments (ref. 72), though in a different context that I would expect (line 488), but do not cite it when explaining how the assessments were conducted (line 298). In line 298, the authors wrongly cite the IUCN Red List itself (ref. 4, incomplete), which informs nothing about the process, but do not cite the IUCN guidelines (ref. 3, 57 and 72). For consistency, I would also recommend avoiding using different versions of the guidelines (ref. 3, 57 and 72) throughout the manuscript – unless they deal with different and specific topics.

3) I suggest using the term ‘endemic’ in a more consistent way throughout the entire manuscript. In many cases, it is not clear if authors mean ‘endemic to Ecuador’ (what it is especially relevant if this is a national assessment) or ‘endemic to a specific region inside Ecuador’.

4) I suggest improving the discussion on the differences between the present and previous assessments. The authors say: “The differences are probably due to broader knowledge, including taxonomic revisions and species descriptions, but also to the different assessment procedures”. This is too superficial and must be better explained.

First thing to note: are both assessments really comparable? Are they both regional/national or global assessments? Did both assessments use data only from inside Ecuador or also from other countries? Did both assessments use the same guidelines? What are the “different assessment procedures” referred by the authors? Which assessment did not follow the IUCN guidelines? This must be specified. Once this is clarified, one can try to understand why there were so many differences.

DD species, for instance. What has changed since the last assessment that allowed the authors to assess 127 species that were previously considered DD? Did the knowledge about these species increased so much over the past 10 years? Or did the definition of 'sufficient' and 'insufficient' 'information to be evaluated' that in fact change? This is especially relevant to note if we take into account that the authors of both assessments are not the same.

Further information and discussion are also needed for taxa that are now listed in higher and lower categories. Again, what explains these differences? Different methods and approaches? This should not be the case, at least in theory, if both assessments followed the IUCN guidelines. Even though there is room for subjectivity and personal judgements in the application of IUCN criteria, this alone should not be enough to explain such big differences between both assessments (139 taxa). Possible explanations for it could be: a) one of the assessments did not (correctly) follow the IUCN guidelines (and that is why it is so important to stress out if and which guidelines were followed in this work); b) scientific knowledge has greatly improved over the past 10 years, allowing for much more refined and detailed assessments; c) over the past 10 years, the overall environmental conditions have improved in some regions (allowing populations/species previously under threat to recover) and decreased in others (threatening populations/species that were in a better situation when the previous assessment was conducted). If this is really the case, it is especially relevant and would be, in my view, the main finding of this study.

Anyway, I guess the most plausible explanation for the differences between both assessments is the combination of all these factors (and maybe others). But authors must discuss this better in the paper and try to provide more detailed explanations. Providing specific examples to illustrate these cases would be especially important.

5) In the section ‘Towards a robust and objective methodology to evaluate species conservation status’, authors suggest their methodology has the potential to improve red list assessments. What are the specific improvements/developments of your methodology with relation to others? How can other evaluators replicate your methodology? In my view, the methods are not described in the manuscript in such a detailed way that it would allow easy and straightforward adoption in other assessments. I acknowledge that the authors have used a robust methodology with the potential to improve future evaluations. But I am not sure if this manuscript, in the way that it is presented, provides enough information on how the methodology can be replicated and used in a standardised way in future assessments. In my view, the methodology is described in a sufficient and adequate way to explain how authors processed their data. This is clear. But, at least to me, it is not clear how they used the data they generated to feed the evaluations following the IUCN guidelines and criteria.

In this sense, I am not sure that it is accurate to say that the methodology is a “key step of improving the protocol for Red List assessment in the effort to validate the taxonomic and spatial database”. If authors really want to keep this, they should make clear: a) to which ‘Red List assessment’ are they referring? To the IUCN ones? If so, please specify (and include references), and point out what exactly are the new things and benefits that the new methodology brings to the table; b) what are the differences, the improvements and advances of the new methodology with relation to the existing ones? c) how exactly the new approach was used when applying the IUCN criteria? d) how can the new methodology be replicated and implemented by other evaluators in other regions and countries?

In fact, if authors are confident about the potential of the new methodology (which I believe they are, and I myself would agree with them), I would suggest writing a second methodological paper, where they describe the methods in a didactic way (i.e. in a guideline format). I would specially recommend contacting the IUCN Amphibian Red List Authority to propose a joint paper incorporating the new methodology into the IUCN guidelines. But, to be honest, I don’t consider that the manuscript, as it is currently written, can be used as a model for future evaluations. Not because I don’t consider the methodology robust (which I do), but mainly because it is not described in a way that would allow its implementation by other evaluators. In other words, it is described as “this is how we did” but not as “this is how you should do”. I encourage authors to write a second methodological paper with the “how to do”.

Additional comments and suggestions can be found in the pdf file. Congratulations to the entire team on this great work! The overall quality of the work is impressive. I think now it is just a matter of improving the manuscript a little bit to reflect with more precision the high quality of all the work that was already done.

6. PLOS authors have the option to publish the peer review history of their article (what does this mean?). If published, this will include your full peer review and any attached files.

Reviewer #1: No

---

## [Author Response · Author response to Decision Letter 0]

9 Apr 2021

RESPONSE TO REVIEWERS (HMOA: in Blue)

Dear Dr. Ortega-Andrade,

Thank you for submitting your manuscript to PLOS ONE. After careful consideration, we feel that it has merit but does not fully meet PLOS ONE’s publication criteria as it currently stands. Therefore, we invite you to submit a revised version of the manuscript that addresses the points raised during the review process.

The one non-anonymous referee and myself like the paper and find it makes an important contribution to EC amphibian conservation. The referee suggests

- to strengthen if global or regional/national assessment,

- to better work out the differences between this and earlier assessments,

- that language might need some improvement,

apart from few others aspects (for detials see reviewer comments).

I agree that these things are relatively easy to manage and much improve the paper, which is very good anyway.

● A rebuttal letter that responds to each point raised by the academic editor and reviewer(s). You should upload this letter as a separate file labeled 'Response to Reviewers'.

● A marked-up copy of your manuscript that highlights changes made to the original version. You should upload this as a separate file labeled 'Revised Manuscript with Track Changes'.

● An unmarked version of your revised paper without tracked changes. You should upload this as a separate file labeled 'Manuscript'.

We look forward to receiving your revised manuscript.

Kind regards,

Stefan Lötters

Academic Editor

PLOS ONE

Journal Requirements:

HMOA: Ok. done.

HMOA: Ok. done.

"The authors have declared that no competing interest exist"

We note that one or more of the authors are employed by a commercial company: Independent researcher.

HMOA: We changed to the correct affiliation for SMRJ, MGL and DCJ into the PARG Project, managed by the Ministry of Environmental and Water, Ecuador. 

3.1. Please provide an amended Funding Statement declaring this commercial affiliation, as well as a statement regarding the Role of Funders in your study. If the funding organization did not play a role in the study design, data collection and analysis, decision to publish, or preparation of the manuscript and only provided financial support in the form of authors' salaries and/or research materials, please review your statements relating to the author contributions, and ensure you have specifically and accurately indicated the role(s) that these authors had in your study. You can update author roles in the Author Contributions section of the online submission form.

HMOA: “The funder provided support in the form of salaries for authors [SMRJ,MGL, DCJ], but did not have any additional role in the study design, data collection and analysis, decision to publish, or preparation of the manuscript. The specific roles of these authors are articulated in the ‘author contributions’ section.” In this specific case, the authors worked in a project managed by the Ministry of Environment and Water of Ecuador.

HMOA: This is not that case.

3.2. Please also provide an updated Competing Interests Statement declaring this commercial affiliation along with any other relevant declarations relating to employment, consultancy, patents, products in development, or marketed products, etc. 

HMOA: This is not that case.

 HMOA: This is not that case.

 HMOA. Thanks for this observation, but this is not the case.

4. We note that Figures 1, 4, 6, 7, 8 and S3, S4, S5 in your submission contain map images which may be copyrighted. All PLOS content is published under the Creative Commons Attribution License (CC BY 4.0), which means that the manuscript, images, and Supporting Information files will be freely available online, and any third party is permitted to access, download, copy, distribute, and use these materials in any way, even commercially, with proper attribution. For these reasons, we cannot publish previously copyrighted maps or satellite images created using proprietary data, such as Google software (Google Maps, Street View, and Earth). For more information, see our copyright guidelines: http://journals.plos.org/plosone/s/licenses-and-copyright.

HMOA: All base maps used to make figures 1, 4, 6, 7, 8 and S3, S4, S5 are available to the public from the interactive Map, from the Ministry of Environment and Water- Ecuador. http://ide.ambiente.gob.ec/mapainteractivo/ and Geographic Military Institute from Ecuador (public shapefiles): http://www.igm.gob.ec/work/files/downloads/mapafisico.html that are not copyrighted.

 4.1. You may seek permission from the original copyright holder of Figures 1, 4, 6, 7, 8 and S3, S4, S5 to publish the content specifically under the CC BY 4.0 license. 

 HMOA: All maps are available to the public from the interactive Map, from the Ministry of Environment and Water- Ecuador. http://ide.ambiente.gob.ec/mapainteractivo/ and Geographic Military Institute from Ecuador (public shapefiles): http://www.igm.gob.ec/work/files/downloads/mapafisico.html that are not copyrighted.

4.2. If you are unable to obtain permission from the original copyright holder to publish these figures under the CC BY 4.0 license or if the copyright holder’s requirements are incompatible with the CC BY 4.0 license, please either i) remove the figure or ii) supply a replacement figure that complies with the CC BY 4.0 license. Please check copyright information on all replacement figures and update the figure caption with source information. If applicable, please specify in the figure caption text when a figure is similar but not identical to the original image and is therefore for illustrative purposes only.

HMOA: All maps are available to the public from the interactive Map, from the Ministry of Environment and Water- Ecuador. http://ide.ambiente.gob.ec/mapainteractivo/ and Geographic Military Institute from Ecuador (public shapefiles): http://www.igm.gob.ec/work/files/downloads/mapafisico.html that are not copyrighted.

 

REVIEWER 1.

Manuscript: Red List assessment for amphibian species of Ecuador: a multidimensional approach for their conservation 

Reviewer: Luis Fernando Marin da Fonte (IUCN Amphibian Specialist Group for Brazil, Amphibian Survival Alliance)

In this paper, the authors provide a very detailed and complete assessment of the conservation status of amphibians in Ecuador besides providing the most up to date list of species for the country. It is a very impressive and important work with great potential to benefit the conservation of this group of animals. Besides having a local importance, with the potential to inform public policies, it can also serve as a role model to other countries globally. I congratulate the authors for their work and recommend its publication with minor changes. That said, I have some suggestions to improve the overall quality of the manuscript and think that, before publication, some important issues should be addressed, as following:

1) with regard to the writing, I acknowledge that the authors are not native English speakers and admire their willingness to publish in a foreign language. Since I am also not a native English speaker, I cannot comment much on the language (though I have made some minor suggestions). The introduction reads very well, but the discussion reads a bit difficult sometimes, with too long and often confusing sentences. I suggest using shorter sentences to improve clarity.

HMOA: Ok. Thanks for this advice. The complete text has been reviewed.

2) My main criticism to this paper (though relatively easy to fix) is that it is not clear in the manuscript if this is a global or a regional/national assessment. Moreover, it is a bit confusing what are the ties of this assessment with the IUCN.

HMOA: Yes. We clarified that it is a National assessment along with the text.

Global assessments (published on the IUCN Red List website) take into account the complete geographic range of taxa, while regional assessments take into account occurrence inside a specific region (usually inside country boundaries). Of course, for species endemic to a given country, global assessments will match national assessments, but this is not the case for species shared by different countries. The IUCN provides specific guidelines for regional assessments, but it is not clear if the authors followed them.

HMOA: Ok, thanks. We have detailed the “Guidelines for Application of IUCN Red List Criteria at Regional and National Levels”, cited into the paper (line 280, page 12). 

All things considered, one would imagine this is indeed a national assessment. However, things get a bit confusing when authors say that they have used the ‘complete distributional range’ of species (line 181). This would not be a problem per se, but it is not clear in the manuscript if the records from other countries were validated by experts - in the same way that records from inside Ecuador were validated by Ecuadorian experts in the workshops. Moreover, it is not clear if the search for records outside Ecuador was exhaustive or not (if I got it right, they were gathered only from online databases?). From what I have understood, records from outside Ecuador were not collected and analysed with the same level of accuracy than the records from inside Ecuador. If this is the case, you would have two kind of data: high quality data from inside Ecuador, and possibly incomplete/unreviewed/not validated data from outside Ecuador. If this is a regional/national assessment, this should not be a problem, since the most relevant data for the evaluations are the ones from inside Ecuador. But if this was intended to be a global assessment, then we would have a problem. In this sense, I think it is fundamental to make it clear, both in the title and in the body of the text, that (or if) this is a national assessment. Moreover, it is important to make clear if data from outside Ecuador was validated by experts and if the search for records was exhaustive (in other words, is this data comparable with the data from inside Ecuador?). Finally, it is important to show how exactly the data from outside Ecuador was used, since this is fundamental to determine if this is a global or a regional/national assessment. 

HMOA: We explain in “Methods” that the niche models were reconstructed with records for the complete distributional range of each species, but the red list assessment was restricted nationally for Ecuador. Records from outside Ecuador were analyzed with the same level of accuracy as the records from inside Ecuador, based on literature records, type localities, and digital databases (photographs, etc…). We based this method on paper-like (cited in line 188, page 8) Syfert et al (2014), Using species distribution models to inform IUCN Red List assessments. Biological Conservation, 177:174-184.

Still in this topic, it is not clear if the authors followed the IUCN guidelines or only the IUCN criteria and categories. This might seem perfumery, but it is not. The IUCN guidelines inform how the criteria and categories should be used, and this is even more relevant in regional assessments. The criteria, in their turn, inform if and how the taxa fall under the different threat categories. In this sense, since the criteria and categories are the same for global and regional assessments, but the guidelines are not, it is not informative enough to only say that the assessment followed “IUCN criteria” (lines 68, 298), “IUCN standards” (lines 298, 459) or “IUCN protocols” (line 298), whatever ‘standard’ and ‘protocol’ mean in this context. It is important to know if the assessment followed the IUCN guidelines and, even more important, if it followed the guidelines for regional assessments.

HMOA: Yes- As commented before, we have detailed the “Guidelines for Application of IUCN Red List Criteria at Regional and National Levels”, cited into the paper (line 280, page 12).

Finally, it is important to make clear if this is an official IUCN assessment (i.e. if members of the IUCN Amphibian Red List Authority were present in the workshops and if the assessment was validated by external members of the IUCN Red List Authority). I specifically mention this because in line 330 the authors say “… the IUCN Red List assessment resulted in…”. If this is not an official IUCN assessment, you cannot say that. In this case, I suggest changing for “... the assessment following the IUCN Red List guidelines and criteria resulted...”. Same thing in line 151.

HMOA: Nationally, this red list assessment is officially validated by the Ministry of Environment of Ecuador (Ministerial agreement 069). National Red List Authority participated in the workshops (Diego F. Cisneros-Heredia, National Red List Authority, Stephanie Arellano, Programme Officer, IUCN Regional Office for South America) but currently we are collaborating with International Red List Authority (Kelsey Neam [Programme Officer - Amphibian Red List Authority - IUCN SSC Amphibian Specialist Group], and Jennifer Luedtke Manager of IUCN Red List Assessments - Global Wildlife Conservation Global Coordinator - Amphibian Red List Authority - IUCN SSC Amphibian Specialist Group). As commented before, we have detailed the “Guidelines for Application of IUCN Red List Criteria at Regional and National Levels”, cited in the paper (line 194, page 9). 

I also suggest reviewing the references to all IUCN documents. They are generally incomplete in the list of references and/or cited without criteria in the body of the text. For instance, the authors do cite the IUCN guidelines for regional assessments (ref. 72), though in a different context that I would expect (line 488), but do not cite it when explaining how the assessments were conducted (line 298). In line 298, the authors wrongly cite the IUCN Red List itself (ref. 4, incomplete), which informs nothing about the process, but do not cite the IUCN guidelines (ref. 3, 57 and 72). For consistency, I would also recommend avoiding using different versions of the guidelines (ref. 3, 57 and 72) throughout the manuscript – unless they deal with different and specific topics.

HMOA: Ok. We have reviewed and updated suggestions.

3) I suggest using the term ‘endemic’ in a more consistent way throughout the entire manuscript. In many cases, it is not clear if authors mean ‘endemic to Ecuador’ (what it is especially relevant if this is a national assessment) or ‘endemic to a specific region inside Ecuador’.

HMOA: Ok. We clarified that “endemic” is referred for exclusive Ecuadorian species (line 260, page 149).

4) I suggest improving the discussion on the differences between the present and previous assessments. The authors say: “The differences are probably due to broader knowledge, including taxonomic revisions and species descriptions, but also to the different assessment procedures”. This is too superficial and must be better explained.

First thing to note: are both assessments really comparable? Are they both regional/national or global assessments? Did both assessments use data only from inside Ecuador or also from other countries? Did both assessments use the same guidelines? What are the “different assessment procedures” referred by the authors? Which assessment did not follow the IUCN guidelines? This must be specified. Once this is clarified, one can try to understand why there were so many differences.

HMOA: Ok. We have extended the comparisons between both assessments along the discussion and included the Table 4, to summarize in detail the conservation status changes from the evaluation in 2011 compared with the current evaluation.

DD species, for instance. What has changed since the last assessment that allowed the authors to assess 127 species that were previously considered DD? Did the knowledge about these species increased so much over the past 10 years? Or did the definition of 'sufficient' and 'insufficient' 'information to be evaluated' that in fact change? This is especially relevant to note if we take into account that the authors of both assessments are not the same.

HMOA: To clarifying those statements, we have included the Table 4 into the manuscript, that detail the number of amphibian species that changed their conservation status from the previous Ecuadorian Red List [27]. Diagonal: Taxa that maintained the same conservation category between assessments. Upper diagonal: Taxa that decrease their conservation category; Lower diagonal: Taxa that increase their conservation category

Further information and discussion are also needed for taxa that are now listed in higher and lower categories. Again, what explains these differences? Different methods and approaches? This should not be the case, at least in theory, if both assessments followed the IUCN guidelines. Even though there is room for subjectivity and personal judgements in the application of IUCN criteria, this alone should not be enough to explain such big differences between both assessments (139 taxa). Possible explanations for it could be: a) one of the assessments did not (correctly) follow the IUCN guidelines (and that is why it is so important to stress out if and which guidelines were followed in this work); b) scientific knowledge has greatly improved over the past 10 years, allowing for much more refined and detailed assessments; c) over the past 10 years, the overall environmental conditions have improved in some regions (allowing populations/species previously under threat to recover) and decreased in others (threatening populations/species that were in a better situation when the previous assessment was conducted). If this is really the case, it is especially relevant and would be, in my view, the main finding of this study.

Anyway, I guess the most plausible explanation for the differences between both assessments is the combination of all these factors (and maybe others). But authors must discuss this better in the paper and try to provide more detailed explanations. Providing specific examples to illustrate these cases would be especially important.

HMOA: Thanks for those comments. We have explained and include statements regarding differences in both evaluations. Punctually, we include this phrase into the text (page 19): “Compared to the previous Ecuadorian National Amphibian Red List [27], we add assessments for 144 species and additionally provide a status evaluation for 135 species that were considered DD at that time (Table 4; S3 Table). As a result of our study, 190 maintained the same conservation status, but 302 taxa have changed - 168 species have now been found to qualify in a higher Red List category, while 64 have been assigned to a lower extinction risk category-. The differences are probably due to broader knowledge, including taxonomic revisions of sanitized locality records, new information of species descriptions along the last decade, ecological models for most of the threatened species, database compilation, and evaluation workshops together with IUCN authorities and specialists [3,59,27].”

5) In the section ‘Towards a robust and objective methodology to evaluate species conservation status’, authors suggest their methodology has the potential to improve red list assessments. What are the specific improvements/developments of your methodology with relation to others? How can other evaluators replicate your methodology? In my view, the methods are not described in the manuscript in such a detailed way that it would allow easy and straightforward adoption in other assessments. I acknowledge that the authors have used a robust methodology with the potential to improve future evaluations. But I am not sure if this manuscript, in the way that it is presented, provides enough information on how the methodology can be replicated and used in a standardised way in future assessments. In my view, the methodology is described in a sufficient and adequate way to explain how authors processed their data. This is clear. But, at least to me, it is not clear how they used the data they generated to feed the evaluations following the IUCN guidelines and criteria.

 In this sense, I am not sure that it is accurate to say that the methodology is a “key step of improving the protocol for Red List assessment in the effort to validate the taxonomic and spatial database”. If authors really want to keep this, they should make clear: a) to which ‘Red List assessment’ are they referring? To the IUCN ones? If so, please specify (and include references), and point out what exactly are the new things and benefits that the new methodology brings to the table; b) what are the differences, the improvements and advances of the new methodology with relation to the existing ones? c) how exactly the new approach was used when applying the IUCN criteria? d) how can the new methodology be replicated and implemented by other evaluators in other regions and countries?

In fact, if authors are confident about the potential of the new methodology (which I believe they are, and I myself would agree with them), I would suggest writing a second methodological paper, where they describe the methods in a didactic way (i.e. in a guideline format). I would specially recommend contacting the IUCN Amphibian Red List Authority to propose a joint paper incorporating the new methodology into the IUCN guidelines. But, to be honest, I don’t consider that the manuscript, as it is currently written, can be used as a model for future evaluations. Not because I don’t consider the methodology robust (which I do), but mainly because it is not described in a way that would allow its implementation by other evaluators. In other words, it is described as “this is how we did” but not as “this is how you should do”. I encourage authors to write a second methodological paper with the “how to do”.

HMOA: Thanks for all those valuable comments. Yes, we are currently working on a second paper regarding the “how to do” as guidelines (López et al, in prep). We have also included key references regarding IUCN protocols, participation of authorities from the IUCN in the workshops and methodology design (lines 1947-199, page 9), and detailed data analysis methods. Furthermore, we have included a fourth table, to summarize in detail the conservation status changes from the evaluation in 2011 compared with the current evaluation.

Additional comments and suggestions can be found in the pdf file. Congratulations to the entire team on this great work! The overall quality of the work is impressive. I think now it is just a matter of improving the manuscript a little bit to reflect with more precision the high quality of all the work that was already done.

HMOA: Thanks a lot for all the interesting and valuable comments. We really appreciate your critical [constructive] review.

---

## [Editor Report · Decision Letter 1]

19 Apr 2021

Red List assessment of amphibian species of Ecuador: a multidimensional approach for their conservation

PONE-D-20-34320R1

Dear Dr. Ortega-Andrade,

We’re pleased to inform you that your manuscript has been judged scientifically suitable for publication and will be formally accepted for publication once it meets all outstanding technical requirements.

Kind regards,

Stefan Lötters

Academic Editor

PLOS ONE
---

## [Editor Report · Acceptance letter]

21 Apr 2021

PONE-D-20-34320R1 

Red List assessment of amphibian species of Ecuador: a multidimensional approach for their conservation 

Dear Dr. Ortega-Andrade:

I'm pleased to inform you that your manuscript has been deemed suitable for publication in PLOS ONE. Congratulations! Your manuscript is now with our production department. 

Kind regards, 

on behalf of

Prof. Dr. Stefan Lötters 

Academic Editor

PLOS ONE